# Derivation of human primordial germ cell-like cells in an embryonic-like culture

Sajedeh Nasr Esfahani[1,15], Yi Zheng [1,14,15], Auriana Arabpour[2,3,4,15], Agnes M. Resto Irizarry[1], Norio Kobayashi[1], Xufeng Xue [1], Yue Shao [5], Cheng Zhao[6], Nicole L. Agranonik[4], Megan Sparrow[4], Timothy J. Hunt[4], Jared Faith [4], Mary Jasmine Lara[4], Qiu Ya Wu[4], Sherman Silber[7], Sophie Petropoulos [6,8,9,10], Ran Yang[11], Kenneth R. Chien [11], Amander T. Clark [2,3,4] ✉ & Jianping Fu [1,12,13] ✉

Primordial germ cells (PGCs) are the embryonic precursors of sperm and eggs. They transmit genetic and epigenetic information across generations. Given the prominent role of germline defects in diseases such as infertility, detailed understanding of human PGC (hPGC) development has important implications in reproductive medicine and studying human evolution. Yet, hPGC specification remains an elusive process. Here, we report the induction of hPGC-like cells (hPGCLCs) in a bioengineered human pluripotent stem cell (hPSC) culture that mimics peri-implantation human development. In this culture, amniotic ectoderm-like cells (AMLCs), derived from hPSCs, induce hPGCLC specification from hPSCs through paracrine signaling downstream of *ISL1*. Our data further show functional roles of NODAL, WNT, and BMP signaling in hPGCLC induction. hPGCLCs are successfully derived from eight non-obstructive azoospermia (NOA) participant-derived hPSC lines using this biomimetic platform, demonstrating its promise for screening applications.

Despite its importance in disease such as infertility[1], hPGC specification during early human embryonic development remains an elusive process. The development of PGCs has been extensively studied using mammalian animal models[2,3]. However, there exists a substantial divergence of the mechanisms for PGC specification among mammals[4]. Development of hPGCLCs from hPSCs is promising for studying human germline development. The most commonly used protocols for generating hPGCLCs require conversion of primed hPSCs into either a naïve-like or incipient mesoderm-like state before hPGCLC specification[5,6]. However, in primate embryos PGCs are specified in the peri-implantation epiblast compartment that has exited from the naïve pluripotency[7]. Primate PGCs emerge while pluripotent

[1]Department of Mechanical Engineering, University of Michigan, Ann Arbor, MI 48109, USA. [2]Molecular Biology Institute, University of California, Los Angeles, Los Angeles, CA 90095, USA. [3]Eli and Edythe Broad Center of Regenerative Medicine and Stem Cell Research, University of California, Los Angeles, Los Angeles, CA 90095, USA. [4]Department of Molecular, Cell and Developmental Biology, University of California, Los Angeles, Los Angeles, CA 90095, USA. [5]Institute of Biomechanics and Medical Engineering, Department of Engineering Mechanics, School of Aerospace Engineering, Tsinghua University, 100084 Beijing, China. [6]Department of Clinical Science, Intervention and Technology, Division of Obstetrics and Gynecology, Karolinska Instituet, 14186 Stockholm, Sweden. [7]Infertility Center of St. Louis, St. Luke's Hospital, St. Louis, MO 63017, USA. [8]Département de Médecine, Université de Montréal, Montréal, QC, Canada. [9]Centre de Recherche du Centre Hospitalier de l'Université de Montréal, Axe Immunopathologie, Montreal, QC H2X 19A, Canada. [10]Département de Médecine, Molecular Biology Programme, Université de Montréal, Montréal, QC, Canada. [11]Department of Cell and Molecular Biology, Karolinska Institutet, 171 77 Stockholm, Sweden. [12]Department of Cell and Developmental Biology, University of Michigan Medical School, Ann Arbor, MI 48109, USA. [13]Department of Biomedical Engineering, University of Michigan, Ann Arbor, MI 48109, USA. [14]Present address: Department of Biomedical and Chemical Engineering, Syracuse University, Syracuse, NY 13244, USA. [15]These authors contributed equally: Sajedeh Nasr Esfahani, Yi Zheng, Auriana Arabpour. ✉e-mail: clarka@ucla.edu; jpfu@umich.edu

epiblast cells undergo lineage diversification to engender amniotic ectoderm and gastrulating cells.

In this work, we show that a three-dimensional (3D) biomimetic hPSC culture mimicking the peri-implantation human epiblast development provides an easily implementable culture protocol for deriving hPGCLCs from primed hPSCs (Fig. 1a). This 3D biomimetic culture has an intrinsic amniogenic property, promoting primed hPSCs to differentiate into AMLCs, which in turn drive specification of hPGCLCs from hPSCs in the culture through paracrine induction downstream of *ISL1*. Additionally, we implement the 3D biomimetic culture in multiwell plate formats for screening of hPGCLC induction from eight non-obstructive azoospermia (NOA)

research participant-derived human induced pluripotent stem cell (hiPSC) lines.

## Results

### Derivation of hPGCLCs in a 3D biomimetic culture

In our previous study, we report a Gel-3D culture for primed hPSCs, in which two biophysical factors are modulated: a 3D extracellular matrix (ECM) overlay and an ECM gel bed coated on cover glasses (Fig. 1a; see "Methods"). The original Gel-3D culture is efficient in deriving squamous AMLCs from primed hPSCs, but not for specification of hPGCLCs[8]. We suspected that the Gel-3D culture might be intrinsically amniogenic, or the original Gel-3D culture parameters might constrain

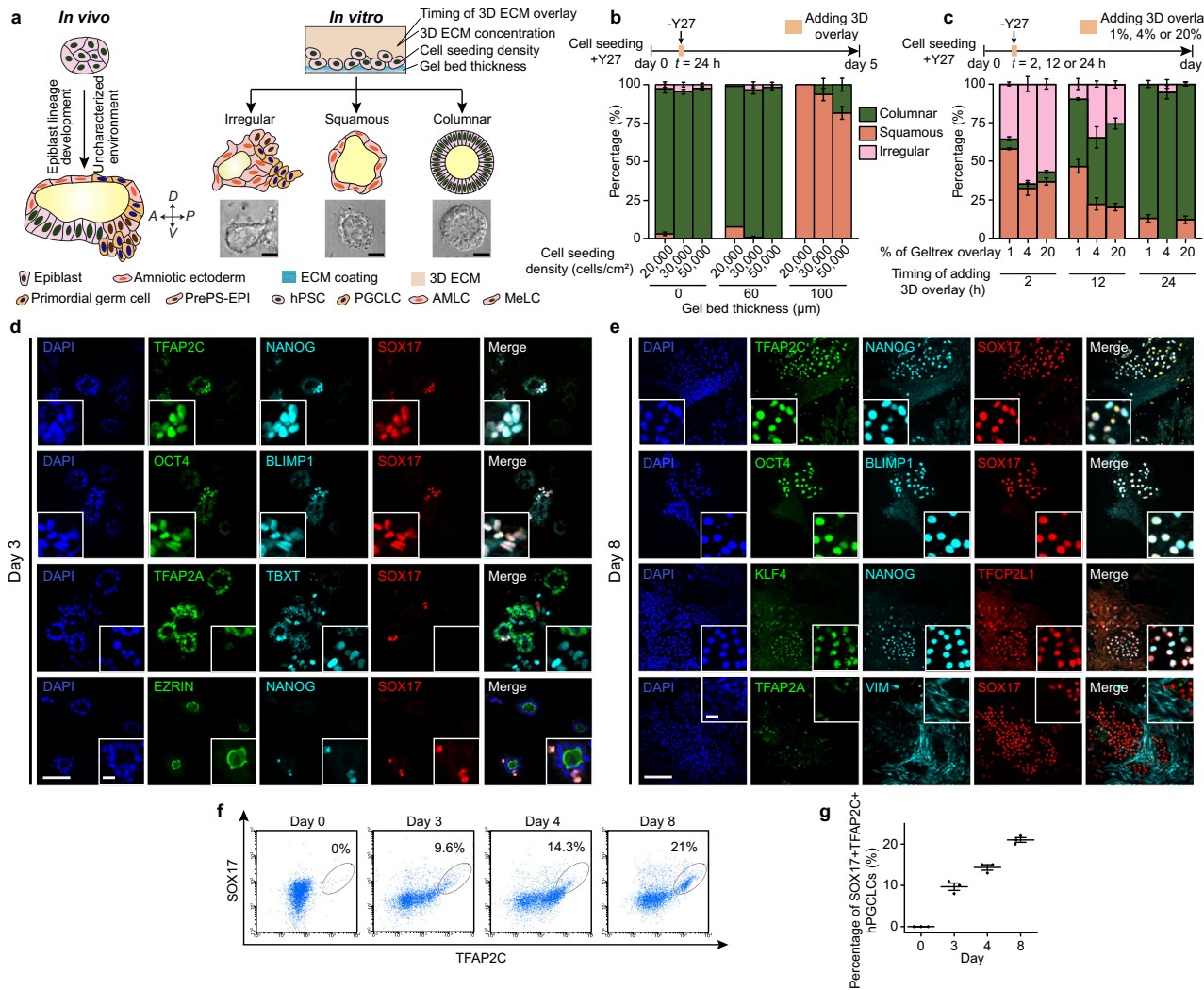

**Fig. 1 | Derivation of hPGCLCs in the Gel-3D culture. a** Schematic of PGC specification in primate embryos (Left). Peri-implantation development of the primate epiblast leads to the formation of the asymmetric embryonic sac, with PGCs emerging in the incipient amniotic ectoderm at the dorsal pole of the embryonic sac and in the gastrulating cells at the posterior end of the embryonic sac. In an in vitro, hPSC-based model of the peri-implantation epiblast development, culture parameters, including gel bed thickness, cell seeding density, 3D ECM concentration, and timing of adding 3D ECM overlay, are modulated to generate columnar epiblast-like cysts, squamous amniotic ectoderm-like cysts, or irregular cysts containing pluripotent and amniotic cells as well as hPGCLCs (Right). Scale bars, 50 μm. **b** Bar plot showing percentages of columnar epiblast-like cysts, squamous amniotic ectoderm-like cysts, and irregular cysts on day 5 as a function of initial cell seeding densities and gel bed thicknesses. Note that 3D ECM overlay was supplemented on day 1. **c** Bar plot showing percentages of columnar epiblast-like cysts, squamous amniotic ectoderm-like cysts, and irregular cysts formed on day 3 as a function of

Geltrex overlay concentration and timing of adding 3D ECM overlay. The quantifications are based on the cyst structures formed in each condition.
**d** Representative micrographs showing immunostaining for EZRIN, NANOG, and SOX17; TFAP2C, NANOG, and SOX17; OCT4, BLIMP1, and SOX17; TFAP2A, TBXT, and SOX17 on day 3 as indicated. **e** Representative micrographs showing immunostaining for TFAP2C, NANOG, and SOX17; OCT4, BLIMP1, and SOX17; KLF4, NANOG, and TFCP2L1; TFAP2A, VIM, and SOX17 on day 8 as indicated.
**f** Enumeration of TFAP2C+SOX17+ hPGCLCs using flow cytometry at indicated time points. **g** Plot showing percentages of TFAP2C+SOX17+ hPGCLCs at indicated time points. In (**b**, **c**, **g**), data represent the mean ± s.e.m. In (**b**, **c**, **f**, **g**), n = 3 independent experiments. In (**d**, **e**), experiments were repeated >10 times with similar results. Nuclei were counterstained with DAPI. Boxed images show magnified views of selected areas. Scale bars, 200 μm (main panel) and 30 μm (inset). Source data are provided as a Source Data file.

lineage development of primed hPSCs towards AMLCs. Thus, herein we first screened different Gel-3D culture parameters, including gel bed thickness, cell seeding density, 3D ECM overlay concentration, and timing of adding 3D ECM overlay (Fig. 1a and Supplementary Movie 1). In our screening, three distinct types of cysts, namely columnar epiblast-like cysts, squamous amniotic ectoderm-like tissues, and irregular cysts, would emerge in the Gel-3D culture (Fig. 1a). When 4% (v/v) Geltrex overlay was added to the Gel-3D culture on day 1, regardless of cell seeding densities (20,000–50,000 cells cm$^{-2}$) or gel bed thicknesses (0, 60, and 100 μm) used, the majority of primed hPSCs developed into either squamous amniotic ectoderm-like tissues or columnar epiblast-like cysts on day 5, without the development of hPGCLCs (Fig. 1a, b and Supplementary Movie 1). However, when 4% (v/v) Geltrex overlay was supplemented 2 or 12 h after cell seeding, with a cell seeding density of 30,000 cells cm$^{-2}$ and gel bed thicknesses of 0 μm, about 65% or 35% of cell colonies on day 3 developed into irregular cystic structures, respectively, which are morphologically distinct from either squamous amniotic ectoderm-like tissues or columnar epiblast-like cysts (Fig. 1a–c and Supplementary Movie 2).

We conducted immunofluorescence assays to examine cell lineage identities in irregular cystic structures, under Gel-3D with 4% (v/v) Geltrex overlay supplemented 2 h after cell seeding, with a cell seeding density of 30,000 cells cm$^{-2}$ and gel bed thicknesses of 0 μm. On day 3, many irregular cystic structures contain EZRIN+ apical lumens (Fig. 1d), and small clusters of hPGCLCs are evident in these structures, expressing early hPGC markers, including TFAP2C, NANOG, SOX17, OCT4, and BLIMP1 (Fig. 1d). Some cells demarcating lumens of the irregular cystic structures express amniotic ectoderm markers ISL1, TFAP2A/B, and GATA3, suggesting their AMLC identity. There are also small cell clusters in irregular cystic structures expressing TBXT, a primitive streak marker, suggesting gastrulating cells present in the culture (Fig. 1d and Supplementary Fig. 1a, b). On day 8, aggregations of hPGCLCs become more evident in the culture (Fig. 1e). In addition to TFAP2C, NANOG, SOX17, OCT4, and BLIMP1, these hPGCLCs express later stage hPGC markers KLF4 and TFCP2L1 (Fig. 1e). TFAP2A+GATA3+ amniotic cells remain segregated from hPGCLCs, and VIMENTIN (VIM) expression is detected in the culture, presumably from the gastrulating cell lineage (Fig. 1e and Supplementary Fig. 1c–e). Flow cytometry analysis shows the percentage of TFAP2C+SOX17+ hPGCLCs in the Gel-3D culture of around 20% on day 8 (Fig. 1f, g), comparable with hPGCLC derivation efficiency from other protocols[5,6]. The robustness of Gel-3D culture for deriving hPGCLCs was confirmed using additional hPSC lines (Supplementary Fig. 2a–c, f), as well as under different medium conditions for both culturing and differentiation of hPSCs (Supplementary Fig. 2d–f).

## Transcriptome analysis

We applied single-cell RNA-sequencing (scRNA-seq) to examine transcriptome of cell lineages in the Gel-3D culture. The Uniform Manifold Approximation and Projection (UMAP) plots reveal five distinct cell clusters, annotated as hPSC, AMLC1, AMLC2, mesoderm-like cell (MeLC), and hPGCLC, on day 3, and three distinct cell clusters, annotated as AMLC, MeLC, and hPGCLC, on day 8 (Fig. 2a, b). In addition to SOX17, TFAP2C, BLIMP1 (also known as PRDM1), and NANOG, hPGCLCs express NANOS3 on day 3 (Fig. 2c and Supplementary Fig. 3a). On day 8, hPGCLCs express additional hPGC markers, including KLF4 (Fig. 2d and Supplementary Fig. 3b). Genes upregulated in day 8 hPGCLCs relative to day 3 hPGCLCs are enriched for those associated with 'protein localization' and 'regulation of cell migration', whereas genes downregulated in day 8 hPGCLCs relative to day 3 hPGCLCs are enriched for 'mRNA metabolic process', 'chromosome organization', and 'nuclear transport' (Supplementary Fig. 3f). AMLC1 and AMLC2 clusters exhibit different expression levels for amniotic ectoderm markers such as TFAP2A, GATA3, and ISL1 (Fig. 2c). Genes that are upregulated in AMLC2 compared to AMLC1 include markers associated with more

mature amnion, such as IGFBP3 and GABRP (Supplementary Fig. 3c–e, g). Additionally, genes upregulated in AMLC2 relative to AMLC1 are enriched for those associated with 'tissue morphogenesis / embryo development' (Supplementary Fig. 3c–e, g). Genes upregulated in day 8 MeLCs compared to day 3 MeLCs are enriched for those associated with 'blood vessel development' and 'cardiovascular system development' (Supplementary Fig. 4a–c). These data prompted us to conduct subclustering analysis for day 8 MeLCs, revealing four subpopulations annotated as endothelium-like cells, smooth muscle-like cells, cardiac mesenchymal-like cells, and cardiac progenitor-like cells (Supplementary Fig. 4d–f).

To further confirm cell lineage identities in the Gel-3D culture, scRNA-seq data from the Gel-3D culture were integrated with those from the Carnegie Stage (CS) 7 human gastrula[9], in vitro cultured human blastocysts up to day 14[10], pre-implantation human blastocysts[11], and the microfluidic post-implantation amniotic sac embryoid (PASE)[12]. As expected, hPSC, AMLC, MeLC, and hPGCLC clusters from Gel-3D on day 3 and day 5 overlap and share high gene module scores with the epiblast, amniotic/embryonic ectoderm, mesoderm, and PGC clusters from the CS7 human gastrula, respectively (Fig. 2e–i). Based on ontogenic M. fascicularis PGC genes, hPGCLCs from Gel-3D on day 3 show highest correlations with hPGCLCs derived using existing culture protocols on day 2 (Fig. 2j). Correlation coefficients between hPGCLCs derived from Gel-3D and hPGCs in the CS7 human gastrula and between hPGCLCs derived from existing culture protocols and hPGCs in the CS7 human gastrula are comparable (Fig. 2j and Supplementary Fig. 5a). Correlation coefficients between AMLC and MeLC clusters from Gel-3D culture and their in vivo and in vitro counterparts were also calculated, repectively (Supplementary Fig. 5b, c).

## Role of ISL1 in hPGCLC specification

Recent studies of primate embryos identify ISL1 as a marker of pre-gastrulation amniotic ectoderm cells[13,14]. In the Gel-3D culture, ISL1+ cells, presumably incipient AMLCs, emerge as early as 24 h after cell seeding (Fig. 3a). At this point, SOX17+ hPGCLCs are not detectable in the culture (Fig. 3a). At 36 h after cell seeding, SOX17+ISL1− cells, presumably incipient hPGCLCs, become detectable (Fig. 3a). Even though SOX17 is also an endodermal lineage maker, the scRNA-seq data generated from the Gel-3D culture support that there are no endodermal cells in the Gel-3D culture. At 72 h, clusters of SOX17+NANOG+ hPGCLCs appear in close proximity to ISL1+ amniotic cells (Fig. 3a). Thus, in the amniogenic Gel-3D culture, AMLCs appear to develop earlier than hPGCLCs.

An inductive effect of the amniotic ectoderm through paracrine signaling in triggering the gastrulation has recently been shown in cynomolgus monkey embryos and the microfluidic PASE[12,13]. We thus suspected that the presence of AMLCs in the Gel-3D culture might play a role in hPGCLC specification. Indeed, transcriptome data from the Gel-3D culture shows upregulated expression of BMP and WNT molecules in AMLCs (Fig. 3b). To examine the role of AMLCs in hPGCLC specification, an AMLC-hPSC co-culture system was developed. Specifically, hPSCs are treated with BMP4 for 48 h to obtain AMLCs that express amniotic ectoderm markers ISL1, TFAP2A/B, and GATA3 (Fig. 3c and Supplementary Fig. 6a, b). These AMLCs can maintain their amniotic cell fate even when cultured for 8 days in mTeSR without exogenous BMP4 supplementation (Fig. 3c and Supplementary Fig. 6a, b). To establish the AMLC-hPSC co-culture, clusters of undifferentiated hPSCs are added to AMLC cultures and cultured in mTeSR for another 2 days. Without any exogenous signals, the presence of AMLCs triggers a pluripotency exit of primed hPSCs and promotes their specification into hPGCLCs within 48 h of co-culture. In the absence of AMLCs, no hPGCLCs is evident in hPSC clusters (Fig. 3c, d and Supplementary Fig. 6b, c). Furthermore, the efficiency of hPGCLC induction within hPSC clusters increases with

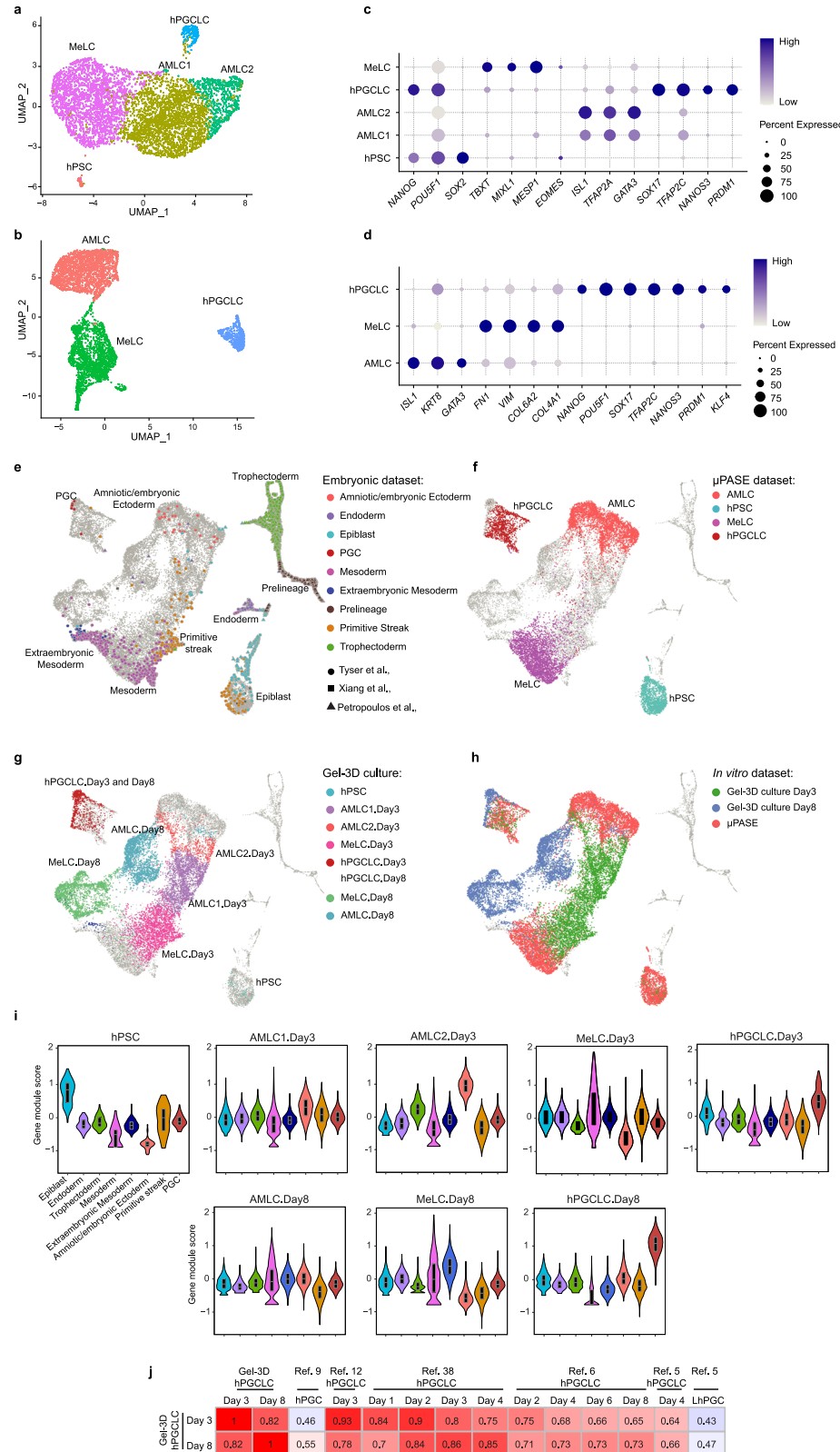

the number of AMLCs present in the co-culture (Supplementary Fig. 6d, e).

Recent studies of mutant *ISL1 cynomolgus* monkey embryos show reduced BMP4 signaling from the amniotic ectoderm, leading to a failure in mesoderm formation during the gastrulation[13]. Given the importance of BMP signaling in PGC specification[7], we next investigated the role of *ISL1* in hPGCLC differentiation in the Gel-3D culture.

Immunofluorescence analysis and quantitative data show drastic reductions of hPGCLC numbers on both day 3 and day 8 in Gel-3D cultures generated from *ISL1*-knockout (KO) hPSCs as compared to wild-type controls (Fig. 3e, f). We further conducted AMLC-hPSC co-culture assays using *ISL1*-KO hPSCs. No hPGCLC specification is detected in wild-type hPSCs when the cells are co-cultured even with a high density of *ISL1*-KO AMLCs (Fig. 3g). In contrast, specification of

**Fig. 2 | Single-cell transcriptomic analysis. a** UMAP plot of scRNA-seq data obtained on day 3, revealing five distinct, color-coded cell populations: hPSC ($n = 56$), AMLC1 ($n = 2254$), AMLC2 ($n = 618$), hPGCLC ($n = 190$), and MeLC ($n = 2087$). **b** UMAP plot of scRNA-seq data obtained on day 8, revealing three distinct, color-coded cell populations: AMLC ($n = 2262$), hPGCLC ($n = 554$), and MeLC ($n = 2245$). **c** Dot plot showing expression of key marker genes in the cell clusters in (**a**) as indicated. **d** Dot plot showing expression of key marker genes in the cell clusters in (**b**) as indicated. **e**–**h** UMAP plot of an integrated scRNA-seq dataset from pre-implantation human embryos[11], in vitro cultured human embryos[10], in vivo human gastrula[9], microfluidic PASEs[12], and day 3 and day 8 Gel-3D cultures. In (**e**), human embryo data are both shape- and color-coded according to the sources and original cell annotations, and microfluidic PASE and Gel-3D data are shown in gray. In (**f**), microfluidic PASE data are color-coded according to original cell annotations. In (**g**), data from day 3 and day 8 Gel-3D cultures are color-coded according to cell annotations in (**a**, **b**). **i** Gene-module score plots for hPSC, MeLC, AMLC1, AMLC2, and hPGCLC clusters identified in the Gel-3D culture. Gene-module scores were calculated based on the top 15 up-regulated genes associated with the epiblast, endoderm, trophectoderm, mesoderm, extraembryonic meso-derm (or ExE_Mesoderm), amniotic / embryonic ectoderm, primitive streak, and PGC from human embryos, including pre-implantation human embryos[11], in vitro cultured human embryos[10], and in vivo human gastrula[9]. **j** Heat map of correlation coefficients among hPGCLCs derived in the Gel-3D culture on day 3 (hPGCLC.Day 3) and on day 8 (hPGCLC.Day8), hPGCs from in vivo human gastrula[9], hPGCLCs from microfluidic PASEs[12], hPGCLCs from different in vitro protocols[5,6,38], and gonadal hPGCs (LhPGC[5]). Correlation coefficients were calculated based on ontogenic cynomolgus monkey PGC (CyPGC) genes[7] (447 in common out of 544). In (**i**), box: 25–75%, bar-in-box: median, and whiskers: 1% and 99%.

hPGCLCs is evident in *ISL1*-KO hPSCs when they are co-cultured with wild-type AMLCs (Supplementary Fig. 7a). While *ISL1*-KO hPSCs can give rise to AMLCS in both Gel-3D and 2D cultures (Supplementary Fig. 7b, c), there are notable reductions of *BMP2/4* in *ISL1*-KO AMLCS compared to wild-type AMLCs (Fig. 3h and Supplementary Fig. 7d). To further examine the roles of BMP and WNT signaling, either BMP4 or WNT3A, or both morphogens were added to Gel-3D cultures developed from *ISL1*-KO hPSCs, respectively, with data showing partial rescues of hPGCLC induction under these conditions (Fig. 3i, j). The effects of supplementing WNT3A, BMP4, or both on rescuing hPGCLC specification from *ISL1*-KO hPSCs were further confirmed in AMLC-hPSC co-cultures (Supplementary Fig. 7e, f).

### Signaling pathways involved in hPGCLC specification

We further conducted drug perturbation assays using IWP2, an inhibitor of WNT ligand secretion, IWR1, an inhibitor specifically targeting the turnover of AXIN2 (a member of the β-catenin destruction complex), XAV9393, an inhibitor of tankyrase enzymes (a family of enzymes involved in the regulation of WNT/β-catenin signaling pathway), and LDN 193189, an inhibitor of ALK2/3 receptors that bind to BMP2/4/7, at different time points during Gel-3D cultures. All the drug treatments during either day 1 or day 2 of Gel-3D cultures significantly inhibit hPGCLC specification by day 3 (Supplementary Fig. 8). The impacts of supplementing IWP2, IWR1, XAV939, and LDN 193189 on hPGCLC specification from hPSCs were further corroborated in AMLC-hPSC co-culture assays (Supplementary Fig. 9a). We conducted additional control experiments to verify that these drug inhibitors do not alter AMLC identity in the co-culture assays (Supplementary Fig. 9b). Together, our data suggest paracrine induction from AMLCs downstream of *ISL1*, likely involving WNT and BMP signals, is critical for hPGCLC specification; however, *ISL1* per se is not required for hPGCLC specification. It should be noted, however, that the drug inhibition assays cannot exclude the possible effects of drug inhibitors on autocrine signaling during fate transition from hPSCs to incipient hPGCLCs.

In addition to BMP and WNT signals, NODAL signaling has also been reported important in PGC specification[15–19]. Accordingly, we characterized the role of NODAL signaling in the Gel-3D culture. Violin plots of scRNA-seq data from day 3 Gel-3D culture show upregulated expression of *LEFTY2* and *NODAL*, two NODAL target genes, as well as of *SMAD2* and *SMAD3*, downstream effectors of NODAL signaling, in hPGCLCs (Supplementary Fig. 10a). Immunofluorescence analysis of day 3 Gel-3D culture also revealed nuclear localization of phosphorylated SMAD2/3 (pSMAD2/3) in hPGCLCs (Supplementary Fig. 10b). Using a *NODAL*-KO hPSC line in Gel-3D cultures resulted in the development of much less hPGCLCs on day 3 (Supplementary Fig. 10c). Nonetheless, supplementing ACTIVIN A in Gel-3D cultures with *NODAL*-KO hPSCs rescued hPGCLC development on day 8 (Supplementary Fig. 10d–f). RT-PCR analysis of day 3 Gel-3D cultures of *NODAL*-KO hPSCs confirmed downregulated expression of hPGC

markers *TFAP2C* and *SOX17* (Supplementary Fig. 10g). We also examined the impact of supplementing SB 43154, a TGF-β receptor inhibitor, in both Gel-3D cultures and AMLC-hPSC co-cultures, with data showing an absence of hPGCLC specification under both culture conditions (Supplementary Fig. 10h, i).

Our drug perturbation assays in Supplementary Figs. 8 and 9 suggest that the hPGCLC specification program has initiated during the first two days of Gel-3D culture. To corroborate this, we conducted live imaging and lineage tracing for the first 3 days of Gel-3D cultures (see "Methods"). In order to track individual cells, live-cell videos were analyzed using a CNN (convolutional neural network)-LSTM (long short-term memory) machine learning classifier [CNN-LSTM][20]. The hPGCLC or non-hPGCLC identity of cells in Gel-3D cultures at $t = 70$ h was ascertained by immunostaining for hPGCLC markers TFAP2C and SOX17. Supplementary Fig. 11a shows two hPGCLCs detected at $t = 70$ h and their lineage history. For cell divisions that give rise to one hPGCLC daughter cell and another non-hPGCLC daughter cell, we could assume that parental cells have not yet fully committed their fates into hPGCLCs. A quantification of cell divisions resulting in hPGCLCs is included in Supplementary Fig. 11b. During the period between $t = 24$–48 h, there were cell divisions that producing both one and two daughter cells with the hPGCLC identity. After $t = 48$ h, however, all cell divisions producing hPGCLC daughter cells were ones in which both daughter cells were hPGCLCs. This observation supports that in the Gel-3D culture, commitment into the hPGCLC fate is established before $t = 48$ h and that parental cells that give rise to two hPGCLC daughter cells are themselves fully committed hPGCLCs.

### hPGCLC induction using hiPSCs from NOA patients

The Gel-3D culture can be easily implemented in multiwell plate formats, which is a useful feature for evaluating multiple hPSCs in parallel. In addition, the Gel-3D culture is compatible with standard immunofluorescence microscopy, ideal for imaging-based screening applications. In comparison, conventional protocols for generating hPGCLCs rely on floating aggregate cultures, which requires whole-mount staining or lineage reporter lines to obtain quantitative data[21,22]. To demonstrate the utility of Gel-3D culture for imaging-based screening applications with multiple hPSC lines in parallel, we sought to conduct screening assays of hPGCLC induction from hiPSCs derived from eight research participants diagnosed with NOA. These hiPSC lines (NOA1, 3, 5, 6, 8–11) were consented for biomedical research, including infertility research and germ cell differentiation (Supplementary Fig. 12). NOA, defined as no sperm in the ejaculate due to failure of spermatogenesis, is the most severe form of male infertility[23,24]. Screening assays with NOA patient-derived hiPSCs demonstrate that the Gel-3D culture can be applied to both established hPSC lines and patient-derived hiPSC lines. Furthermore, they support the potential of the Gel-3D culture to be widely adopted by the research community for hPGCLC induction assays and translational studies when evaluating multiple cell lines in parallel. It should also be noted that it remains unclear whether NOA

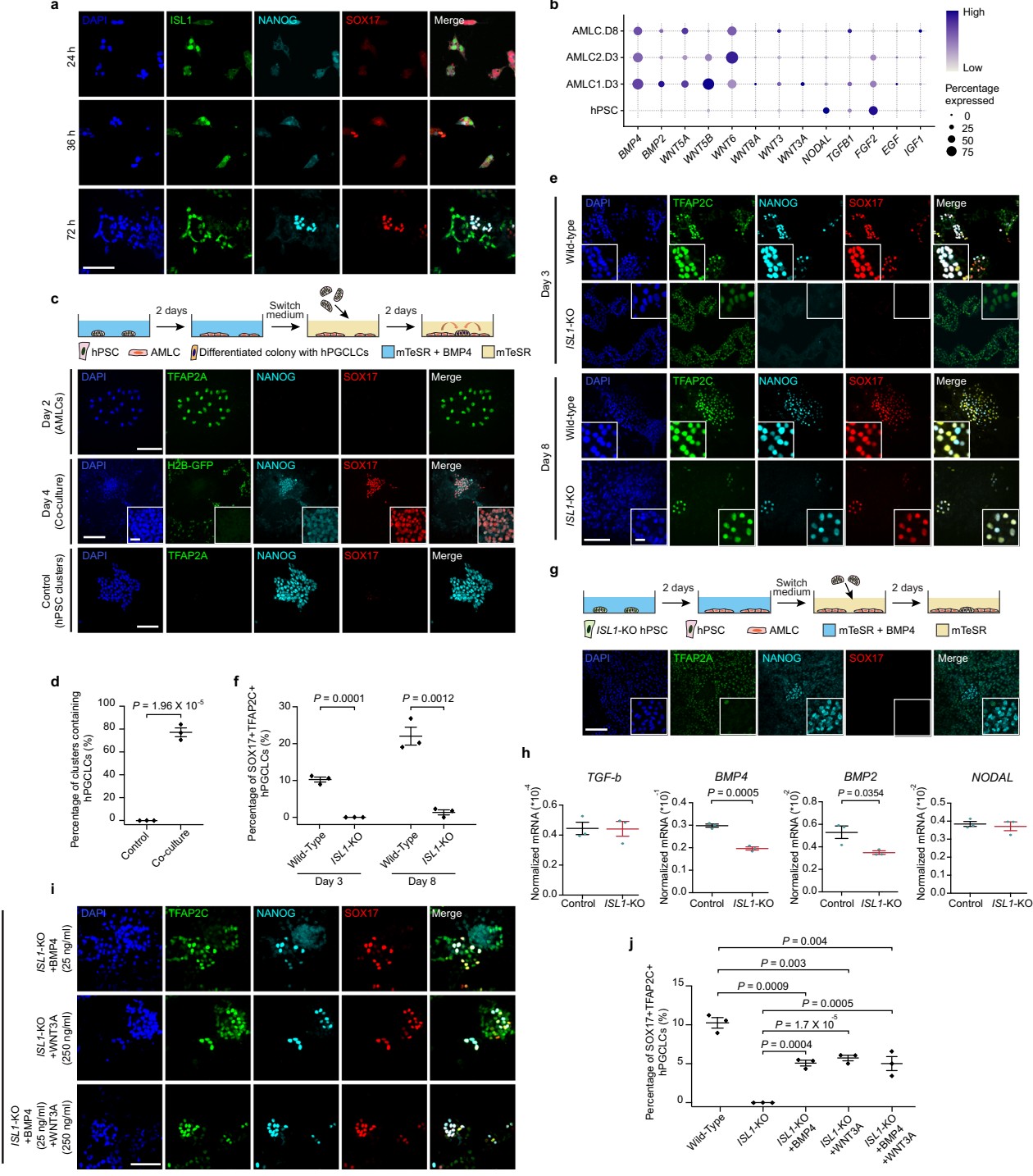

**Fig. 3 | Amniotic ectoderm-like cells induce hPGCLC specification in the Gel-3D culture involving ISL1. a** Micrographs showing immunostaining for ISL1, NANOG, and SOX17 at indicated time points. **b** Dot plot showing expression of selected signaling factors in AMLCs and hPSCs as indicated. **c** Co-culture assays of AMLCs and hPSCs. Micrographs show immunostaining on day 2 (AMLCs) for TFAP2A, NANOG, and on day 4 (co-culture) for NANOG and SOX17, and for hPSC clusters without AMLCs as a control for TFAP2A, NANOG, and SOX17 as indicated. **d** Plot showing percentages of clusters containing NANOG+SOX17+ hPGCLCs in the co-culture system. **e** Micrographs showing immunostaining of the Gel-3D culture with *ISL1*-KO and wild-type hPSC lines. Cells were stained for TFAP2C, NANOG, and SOX17 on day 3 and day 8 as indicated. **f** Plot showing percentages of TFAP2C +SOX17+ hPGCLCs at indicated conditions. **g** Co-culture assays of *ISL1*-KO AMLCs and hPSCs. Micrographs show immunostaining for TFAP2A, NANOG, and SOX17 on

day 4. **h** RT-PCR analyses showing expression of *BMP2, BMP4, TGF-β, and NODAL* in the AMLCs generated in 2D on day 2 from wild-type and *ISL1*-KO hPSCs as indicated. **i** Micrographs showing immunostaining of the Gel-3D culture with *ISL1*-KO hPSCs supplemented with BMP4 (25 ng ml$^{-1}$), WNT3A (250 ng ml$^{-1}$), or the combination of BMP4 and WNT3A as indicated. Cells were stained for TFAP2C, NANOG, and SOX17 on day 3. **j** Plot showing percentages of TFAP2C+SOX17+ hPGCLCs at indicated conditions. In (**a, c, e, g, i**), experiments were repeated three times with similar results. Nuclei were counterstained with DAPI. Boxed images show magnified views of selected areas. Scale bars, 200 μm (main panels) and 30 μm (insets). In (**d, f, h, j**), $n = 3$ independent experiments, and data represent the mean ± s.e.m. $p$ values were calculated using unpaired, two-sided Student's $t$ test. Source data are provided as a Source Data file.

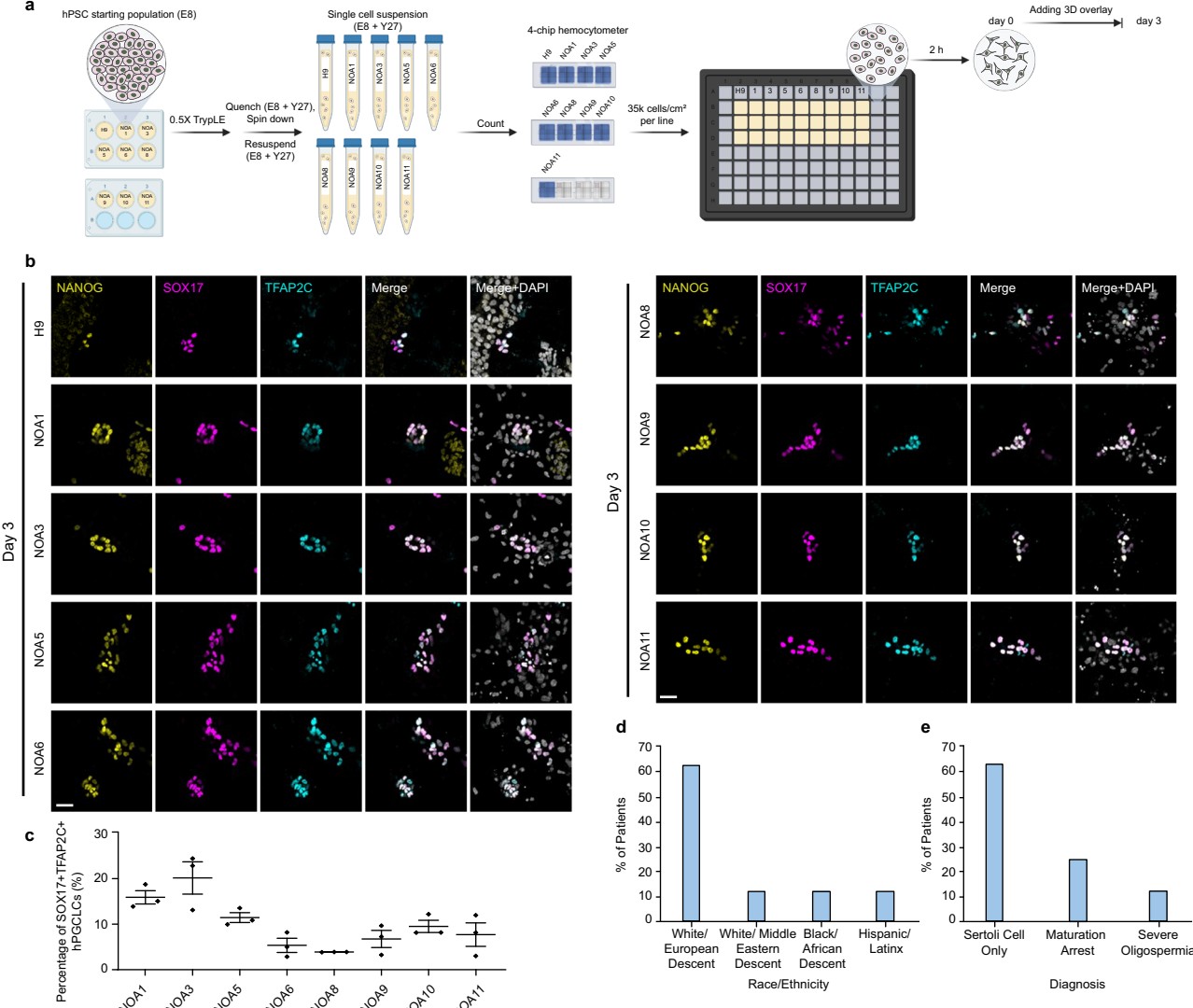

**Fig. 4 | Gel-3D culture for screening research participants diagnosed with NOA.** **a** Schematic of screening protocol developed to evaluate hPGCLC induction from eight non-obstructive azoospermia (NOA) research participant-derived hPSC lines (NOA1,3,5,6,8–11). The hPSC line H9 was used as a positive control. Schematic was made with BioRender.com. **b** Micrographs showing immunostaining for NANOG, SOX17, and TFAP2C on day 3 for different hiPSC lines, as indicated. **c** Plot showing percentages of TFAP2C+SOX17+ hPGCLCs induced from eight NOA research participant-derived hPSC lines as indicated in the Gel-3D culture system on day 3. **d**, **e** Bar plots showing races and ethnicities of research subjects and their diagnoses. In (**b**), experiments were repeated twice with similar results. Nuclei were counterstained with DAPI. Scale bars, 30 μm. In (**c**), *n* = 3 independent experiments, and data represent the mean ± s.e.m. Source data are provided as a Source Data file.

patient-derived hiPSCs could give rise to hPGCLCs. Specifically, hiPSCs derived from the eight NOA research participants were cultured in 96-well plates using the Gel-3D culture (Fig. 4a). On day 3, clusters of hPGCLCs co-expressing TFAP2C, NANOG, and SOX17 are evident from all the eight NOA participant-derived hiPSCs, suggesting that hiPSCs from NOA participants retain the ability to give rise to hPGCLCs (Fig. 4b–e). For the equitable use of new tools and technologies in biomedical research[25] in this study we included NOA participants of different ancestries (Fig. 4d).

## Discussion

In summary, in this work we have successfully developed an easily implementable but highly efficient Gel-3D culture for deriving hPGCLCs from hPSCs. We show that the Gel-3D culture has an intrinsic amniogenic property, promoting primed hPSCs to differentiate into AMLCs, which in turn drive specification of hPGCLCs in the culture through paracrine induction downstream of *ISL1*. Using genetic and drug perturbation assays, we further show the roles of

NODAL, WNT, and BMP pathways in hPGCLC specification in the Gel-3D culture. Since primate PGCs emerge together with amniotic ectoderm and gastrulating cells in vivo, the Gel-3D culture provides a more in vivo-like developmental environment that might be useful for dissecting molecular and genetic mechanisms underlying hPGC specification. The data presented in this work also suggest that the timing of adding Geltrex overlay and its concentration are important factors of the Gel-3D culture to establish an amniogenic environment. Importantly, this amniogenic environment remains conductive for hPSCs to choose the lineage differentiation path towards hPGCLCs, under paracrine inductive effects from nascent AMLCs. We have successfully implemented the Gel-3D culture in multiwell plate formats for screening of hPGCLC induction from eight NOA research participant-derived hiPSC lines. Given its in vivo relevance, simplicity, robustness, and compatibility with live imaging and screening applications, the Gel-3D culture represents a promising tool for deriving hPGCLCs from hPSCs and studying the mechanisms underlying their origin and specification.

## Methods

### Cell lines

Human pluripotent stem cell (hPSC) lines used in this study include H9 human embryonic stem cell (hESC; WA09, WiCell; NIH registration number: 0062), H1 hESC (WA01, WiCell; NIH registration number: 0043), HES-3 hESC (WiCell), 1196a line (human induced pluripotent stem cell, or hiPSC, from the University of Michigan Pluripotent Stem Cell Core[26]), and eight hiPSC lines derived from research participants diagnosed with NOA. All protocols for using these hPSC lines have been approved by the Human Pluripotent Stem Cell Research Oversight Committee at the University of Michigan or in the case of the NOA hiPSC lines, the UCLA human pluripotent stem cell research oversight (hpSCRO) committee, and the UCLA Office of Human Research Oversight Protection Institutional Review Board as described below. All hPSC lines have been authenticated by the original sources as well as in-house by immunostaining for pluripotency markers and successful differentiation to the three germ layer lineages. All hPSCs were maintained in a feeder-free culture for at least 10 passages and authenticated as karyotypically normal at the indicated passage number. Karyotype analysis was performed by Cell Line Genetics or in the case of the NOA research participant lines, KaryoStat Assays (Thermo Fisher). All hPSC lines were tested negative for mycoplasma contamination (LookOut Mycoplasma PCR Detection Kit, Sigma-Aldrich).

### Derivation of hiPSC lines

Disease-specific hiPSC lines were generated from research participants diagnosed with nonobstructive azoospermia (NOA). This study was approved and reviewed annually by the UCLA Institutional Review Board (IRB #18-001466) together with annual review by the UCLA Human Pluripotent Stem Cell Research and Oversight (hPSCRO) Committee (hPSCRO #2018-005-04). Research subjects consented to a biopsy to generate hiPSCs for differentiation of germ cells. Following the procedure, a 1-mm skin punch biopsy was shipped overnight on cold packs to UCLA in DMEM/F12 (Thermo Fisher Scientific, # 11-320-082). Upon receipt at UCLA, the biopsy was digested with collagenase IV (Life Technologies, #17018029) for 1 h at 37 °C and 5% $CO_2$. Digested tissues were plated on tissue culture plates coated with 0.1% gelatin (Sigma-Aldrich, # G1890-100G) in a fibroblast culture medium consisting of 15% fetal bovine serum (FBS; GE Healthcare # 26140079), 1% non-essential amino acids (Invitrogen, # 11140050), 1% glutamax (Gibco, # 35050061), 1% penicillin-streptomycin-glutamine (Gibco, # 10378016) and primocin (Invivogen, # ant-pm-2). Fibroblast outgrowths from tissue pieces were monitored for 3–4 weeks with the medium refreshed every 3 days. When the tissue culture dish became confluent, fibroblasts were passaged using 0.05% trypsin (Gibco, # 25-200-056) and replated to create human dermal fibroblasts (HDFs) which were banked under liquid nitrogen between passage 2-5. For reprogramming, HDFs were thawed and cultivated in the fibroblast culture medium, as described above. Once HDFs became ~80% confluent, they were infected with sendai virus (SeV) using the CytoTune-iPS 2.0 sendai reprogramming kit (Life Technologies, # A16518) according to manufacturer's instructions. Starting around 3 weeks after infection, individual colonies were picked and expanded either on mitomycin-c treated mouse embryonic fibroblasts (NOA1) in hiPSC medium, DMEF/F-12, 20% knockout serum replacement (Life Technologies, # 10-828-028), 10 ng/mL bFGF (R&D Systems, # 233-FB-010), 1% non-essential amino acids, 1% penicillin-streptomycin-glutamine, primocin, and 0.1 mM β-mercaptoethanol (Sigma-Aldrich, # 21985-023)) or picked directly into Essential 8 medium (E8, Thermo Fisher Scientific, # A1517001) A1517001 and cultured on vitronectin (VTN-N, ThermoFisher Scientific, # A14700) coated tissue culture plates (NOA3,5,6,8-11). VTN-N is a recombinant human protein as truncated basement coating. NOA1 was transitioned to E8 and VTN-N by passage (p) 12. Fibroblasts and resulting hiPSC lines were karyotyped using

karyoStat™ (Thermo Fisher Scientific,) and authenticated using Cell ID report (Thermo Fisher Scientific). Self-renewal property of hiPSCs was verified using immunofluorescence for NANOG, OCT4, SSEA4, and TRA-1-91 as previously described[27]. Additional information on research participants' fibroblast and hiPSCs (NOA1,3,5,6,8-11) are listed in Supplementary Table 3. All hPSC lines tested negative for mycoplasma contamination (MycoAlert Detection Kit, Lonza, # LT07-318).

### Cell culture

hPSCs were maintained in a standard feeder-free culture systems using mTeSR medium (mTeSR; STEMCELL Technologies, # 85850) or TeSR-E8 medium (Essential 8 or E8; STEMCELL Technologies, # 05990). Lactate dehydrogenase-elevating virus (LDEV)-free, hESC-qualified reduced growth factor basement membrane matrix Geltrex (Thermo Fisher Scientific; derived from Engelbreth-Holm-Swarm tumors similarly for Matrigel, # A1413302) was used in Gel-3D culture. For NOA research participant-derived hiPSC lines, VTN-N basement coating were used for coating tissue culture plates. Experiments using the NOA hiPSC lines were conducted between P7-30. For the rest of the cell lines the hPSCs were used before reaching P70.

### Generation of Gel-3D culture

Cultured hPSC colonies were first incubated with Accutase (Sigma-Aldrich, # A6964) at 37 °C for 10 min. hPSCs were then centrifuged, and the resultant cell pellet was resuspended in mTeSR containing 10 μM Y27632 (Tocris, # 1254), a ROCK inhibitor that prevents dissociation-induced apoptosis[28]. Coverslips were pre-coated with 1% Geltrex at 37 °C for at least 1 h. hPSCs were plated as single cells at 30,000 cells cm$^{-2}$ onto the indicated substrate.

To establish 3D ECM overlay, 2 h after initial cell seeding, the culture medium was changed to fresh mTeSR or Essential 6 medium (ThermoFisher Scientific, # A1516401) containing 4% (v/v) Geltrex and was replenished daily thereafter. Y27632 was removed from culture medium at 24 h after initial cell seeding (day 1).

### Modification of Gel-3D culture for screening

Cultured hPSC colonies were first washed with Dulbecco's phosphate-buffered Saline (D-PBS; Invitrogen, # 14190144) and incubated with 0.5× TrypLE select enzyme (Gibco, # 50-591-419) and 0.5 mol l-EDTA solution (Nacalai USA, pH 8.0, # 13567-84) at 37 °C, 5% $CO_2$ for 10 min. hPSCs were quenched with E8 medium containing 10 μM Y27632 and gently pipetted to dissociate to single cells. Single cell suspension was centrifuged at 300 g for 5 min before resuspending resultant cell pellets gently in E8 medium containing 10 μM Y27632. Cell counts of all hPSCs were done in parallel using 4-chip disposable hemocytometers (Bulldog Products). For hPGCLC induction screen in 96-well plates, Ibidi USA μ-plate 96 well black, ibiTreat–tissue culture treated polymer coverslip, sterilized plates (Ibidi, Fisher Scientific, # 89626) were used, where each well was pre-coated with 1% Geltrex for at least 1.5 h at 37 °C and 5% $CO_2$. Single-cell suspension of each hPSC line was plated in each well at 35,000 cells cm$^{-2}$ in technical triplicate. In total, 2.5 h after initial cell seeding, culture medium (E8 medium containing 10 μM Y27632) was changed to fresh E8 medium containing 4% Geltrex and 10 μM Y27632. Culture medium was replenished daily thereafter. Y27632 was removed from culture medium at 24 h after initial cell seeding.

### Co-culture assay

H2B-GFP hPSCs suspended in mTeSR containing 10 μM Y27632 were plated as single cells at 10,000 cells cm$^{-2}$ onto a coverslip precoated with 1% Geltrex at 37 °C for at least 1 h. In total, 24 h after cell seeding, culture medium was switched to mTeSR supplemented with BMP4 (R&D SYSTEMS, # 314-BP-050, 50 ng mL$^{-1}$), and the cells were cultured for another 48 h. At this point, the culture medium was replaced with fresh mTeSR without BMP4, before small clusters of undifferentiated hPSCs were plated onto the coverslip. Cells were

cultured for 2–8 days in mTeSR without BMP4 before downstream analysis.

### NODAL-knockout hPSCs

NODAL-knockout (KO) clones were generated from H9 hPSCs by deleting a 58-bp portion of genomic DNA within exon 1 by CRISPR/Cas9. Two cRNA purchased from ThermoFisher Scientific [NODAL_crRNA_1: 5′-AGGCUCAGCAUGUACGCCAG-3′; NODAL_crRNA_2: 5′-AGACAUCAUCCGCAGCCUAC-3′] were used for this purpose. A standard protocol was utilized to prepare duplexes of crRNA:tracrRNA and introduce into H9 cells with the Cas9 enzyme and the pCXLE-EGFP expression plasmid for constitutional expression of EGFP using the NEON electroporation system (Thermo Fisher Scientific). EGFP-expressing single cells were collected into Matrigel-coated 96-well plates by fluorescence-activated cell sorting using FACSAria Fusion (BD Biosciences) with the CloneR single-cell culture supplement diluted with mTeSR Plus medium (STEMCELL Technologies, # 100-1130). To detect the anticipated deletion, genomic DNA was isolated from the single-cell derived clones and subjected to PCR using the following primers designed to amplify NODAL exon 1 [Forward Primer: 5′-CTTCCTTCTGCACGCCTGGTGG-3′; Reverse Primer: 5′-CCAACCCA-CAGCACTTCCCGAG-3′]. The resulting amplicons were subjected to direct Sanger sequencing using a primer 5′- CTTCCTTCTGCA CGCCTGGTGG-3′.

### ISL1-knockout hPSCs

HES-3 hPSC line was used to generate ISL1-knockout (KO) cell line by applying CRISPR/Cas9 with the guide RNAs used in the paper[13]. This cell line was a gift from Dr. Kenneth R. Chien.

### Quantification of immunofluorescent images

To quantify the percentage of hPGCLCs in the immunofluorescent images, we used samples from three independent experiments and more than 20 images were captured from different positions of each sample. All immunofluorescent images were taken using the same confocal microscope with the same setups. The images were imported into Fiji software and enhanced using identical parameters. The number of TFAP2C+NANOG+SOX17+ cells was manually counted. DAPI staining was utilized to determine the total cell count. We calculated the percentage of hPGCLCs in each image.

### BMP, WNT and TGF-B inhibition assays

In the small-molecule inhibitor treatment assays, either 500 nM BMP inhibitor LDN193189 (LDN; STEMCELL Technologies, # A1516401), or 5 µM Wnt inhibitor IWP2 (Selleckchem, # S7085), or 10 µM of Wnt inhibitor IWR1 (Selleckchem and STEMCELL Technologies, # NC1319406), or 10 µM XAV939 (Cell Guidance System, # SM38-10), or 10 µM TGF-B inhibitor SB 431542 (STEMCELL Technologies, # 72234) was added to the culture medium at indicated time points. Dimethylsulfoxide (DMSO; Sigma-Aldrich, # D2650) was added to the control groups.

### Rescue assays

In the assays, either 25–100 ng/ml BMP (R&D SYSTEMS, # 314-BP-050), 50–500 ng/ml WNT3A (Fisher Scientific, # 5036WN010), or the combination of both was added to the culture medium in the Gel-3D culture system and co-culture system for the whole culture time. Dimethylsulfoxide (DMSO; Sigma-Aldrich, # D2650) was added to the control groups.

### Immunocytochemistry

hPSCs were fixed in 4% paraformaldehyde (PFA, Thermo Fisher Scientific, # 043368.9 M; buffered in 1× phosphate buffered saline (PBS; Fisher Scientific, # 10-010-072)) at room temperature for 1 h and permeabilized in 0.1% sodium dodecyl sulfate (SDS, Fisher Scientific,

#AAJ60015AC) for 30 min. Samples from disease-specific hiPSC lines were blocked in 10% donkey serum (Sigma-Aldrich, # S30-100ML) for 1.5 h at room temperature, followed by incubation with primary antibody solutions in 10% donkey serum at 4 °C for 12–16 h. Samples were then labeled with donkey-raised secondary antibodies in 4% donkey serum at room temperature for 45 min. Other samples were blocked in 2% donkey serum at 4 °C for 24 h, followed by incubation with primary antibody solutions in 2% donkey serum at 4 °C for another 24 h. Samples were then labeled with donkey-raised secondary antibodies (1:400 dilution) in 2% donkey serum at 4 °C for another 24 h. Cell nuclei were stained with 4,6-diamino-2-phenylindole (DAPI; ThermoFisher Scientific, # D1306). All primary antibodies, their sources, and dilutions are listed in Supplementary Table 1.

### RT-PCR analysis

To prepare samples for RNA isolation, the samples were first washed with DMEM/F12 to remove Geltrex overlay. A RNeasy Micro Kit (QIAGEN, # 74004) was used following the manufacturer's instructions to isolate RNA from the cell pellet. A NanoDrop 1000 spectrophotometer (Thermo Fisher Scientific) was utilized to determine RNA quality and quantity. Reverse transcription was performed following the iScript cDNA Synthesis Kit (Bio-Rad). RT-PCR analysis was performed using Quantitect Sybr Green MasterMix (QIAGEN, # 330520) and specific primers on a CFX Connect Real-Time System (Bio-Rad).

To quantify relative gene expression, the 2−ΔCt method was used, and human GAPDH was considered an internal control. All analyses were performed with at least three biological replicates and three technical replicates. The investigator who conducted the RT-PCR analysis was blinded to the test condition allocation during the experiment. Primers are listed in Supplementary Table 2.

### scRNA-seq and data analysis

Gel-3D culture at day 3 and day 8 were washed with DMEM/F12 and treated with Accutase for 30 min to obtain single-cell suspensions. Cells from two wells of a 24-well plate were centrifuged at $300 \times g$ for 5 min and transferred into a 1.5 ml tube containing PBS containing 0.5% BSA (Thermo Fisher Scientific, # 11020021). Then the cell suspension is filtered using a 40 um Cell Strainer (Sigma-Aldrich, # BAH136800040) to remove cell clusters. Within 1 h after cell dissociation, cells were loaded into the 10x Genomics Chromium system. 10x Genomics libraries were prepared by the University of Michigan sequencing core according to the manufacturer's instructions. Libraries were then sequenced with coverage of ~20,000 raw reads per cell on an Illumina NovaSeq with paired-end sequencing. scRNA-seq data were aligned and quantified using Cell Ranger Single-Cell Software Suite (v.3.1.0, 10x Genomics) against GRCh38-3.0.0.

The dimensionality reduction and clustering of scRNA-seq data were performed using R package Seurat (v. 3.1.5)[29,30]. Default setups were used unless noted otherwise. In brief, cells with nfeature_RNA ≤ 3000 or ≥5200 for day 3, and ≤1800 or ≥4500 for day 8, or cells in which the total mitochondrial gene expression exceeded 8% of total gene expression were discarded from the analysis. A gene expressed in at least 3 cells was retained for analysis. Gene expression was calculated by normalizing the raw count by the total before being multiplied by 10,000 and log transformed. After cell-cycle regression, principal component analysis was performed using the RunPCA function in Seurat. Identification of cell clusters by a shared nearest neighbor (SNN) modularity optimization-based clustering algorithm was achieved using the FindClusters function. Dimensionality reduction was achieved using Uniform Manifold Approximation and Projection (UMAP) algorism (RunUMAP function). Differentially expressed genes (DEGs) were identified using FindAllMarkers, with a minimal fold difference of 0.25 in the logarithmic scale and >25% detection rate in either of the two cell types under comparison. GO analyses were performed using DAVID Bioinformatics Resources 6.8 based on DEGs (Supplementary Data 1).

For comparison with published data, gene expression data obtained from different platforms (GEO repository, NCBI) were first transformed into log2(reads per million mapped reads (RPM) + 1). The average expression level of each cell type was used for the calculation of the correlation coefficient. The ontogenic gene list selected for calculating the correlation coefficient is critical for the accuracy of cell fate identification. Thus, for hPGCLCs and primitive streak lineages were obtained from previous publications[7,31], while ontogenic gene list was obtained by merge DEGs of early amnion vs. EPI and late amnion vs. early amnion generated using the dataset reported previously[13].

Scatterplot has been generated using log2(RPM + 1) values. Cells and genes were filtered using the same criteria as for all other scRNA-seq analysis. The highly expressed genes were defined as genes whose log2(RPM + 1) values were at least 4-fold greater in one of the cell populations.

### Pre-processing single-cell data and gene expression quantification

Published datasets used in this study were downloaded from GSE134571[12], E-MTAB-3929[11], GSE136447[10], and E-MTAB9388[9]. scRNA-seq data were processed using the Cell Ranger pipeline (v3.0.0) with default parameters, which uses the STAR aligner (v2.5.1b)[32] to map reads to GRCh38 reference genome (v.3.0.0, GRCh38, downloaded from the 10x Genomics website). Published Smart-Seq2 datasets were also mapped on the same reference using the same aligner with default settings to minimize platform and processing differences. Only uniquely mapped reads were kept for gene expression quantification. Raw read counts were further estimated using rsem-calculate-expression from RSEM tool[33] with the option of "--single-cell-prior."

### Quality control and normalization

Cut-off based on the number of expressed genes (nGene) and percentage of mitochondrial genes (percent.mito) were used to filter out the low-quality cells. Low-quality cells from Smart-Seq2 datasets were filtered out by having less than 2000 nGene or percent.mito higher than 0.125. scRNA-seq data were filtered by using the same cut-off reported previously (around 3200 <nGene <6400, percent.mito <0.06)[12]. Cells belong to hemogenic endothelial progenitors and erythroblasts from Carnegie Stage 7 were excluded in the following analysis. After quality control and excluding mitochondria genes, we focused on genes with one or more counts in at least five cells (assessed for each dataset separately) and calculated log-normalized counts using the deconvolution strategy implemented by the computeSumFactors function in R scran package (v.1.14.6)[34] and followed by rescaled normalization using the multiBatchNorm function in the R batchelor package (v.1.2.4)[35]. So, the size factors were comparable across batches. Log-normalized expression after rescaling was further used in integration and marker-gene detection.

### Integrated analysis of multiple datasets

We integrated all datasets based on their mutual nearest neighbors (MNNs) using the RunfastMNN function wrapped in R SeuratWrappers package (v.0.3.0) (https://github.com/satijalab/seurat-wrappers). In detail, it was done by performing a principal component analysis (PCA) on the top 2500 highly variable genes selected by RunFastMNN function and then correcting the principal components (PCs) according to their MNNs. We selected the corrected top 25 PCs for downstream UMAP dimensional reduction using the RunUMAP function in the R Seurat package[36]. As previously discussed[37], cells labeled as "ICM" and "PSA-EPI" from Xiang et al. were excluded in UMAP visualization and marker gene detection because those cells were potentially misclassified in the previous annotation.

### Marker gene detection and calculating gene-module score

According to the previously published annotation, cells from embryonic datasets were divided into 8 major groups (epiblast, endoderm, trophectoderm, mesoderm, extraembryonic mesoderm, amniotic/embryonic ectoderm, primitive streak (without PGCs), and PGCs). FindMarkers function from R package Seurat performed paired-wise differential expression analysis using 'roc' test between groups. The top 15 up-regulated marker genes with at least average power of more than 0.4 conserved in all comparisons were selected to calculate the gene-module score, equal to the mean scaled normalized expression values of top marker genes in each cell. Any scaled values more than 2.5 (less than −2.5) were set to 2.5(−2.5) to avoid the influence of outliers.

### Live cell video acquisition

The micropattern array was imaged using the Zeiss Axio Observer Z1 inverted epifluorescence microscope enclosed in the XL S1 incubator (Carl Zeiss MicroImaging) to maintain cell culture at 37 °C and 5% CO2. Fluorescence images were recorded with a 10x objective for 72 h. A GFP filter set was used for the fluorescent imaging of the nuclei of H2B-GFP cells. The fluorescent imaging was performed using an exposure time of 200 ms and a time frame of 15 min to minimize phototoxic effects on cells.

### Detection of hPGCLCs after live cell imaging

The sample was fixed, permeabilized, blocked, and stained for NANOG and SOX17 to detect hPGCLCs in the Gel-3D culture.

### Live cell video analysis

Live cell video analysis was conducted with a Python pipeline developed by Resto-Irizarry et al.[20]. In brief, individual cells in a time point were identified after (1) contrast enhancement, (2) thresholding, and (3) cell cluster segmentation. Cells were tracked from one time point to another, using Euclidian distance. Division events were identified with a CNN-LSTM (convolutional neural networks-long short-term memory) machine learning classifier and parent and daughter IDs were stored. The final video frame was compared to the stained samples in order to establish cell identity. For cells identified as hPGCLCs, cell position and division data were used for lineage tracing.

### Flow cytometry

Samples from different days were treated with Accutase for 1 h to obtain single-cell suspensions. Single cells were then fixed in 4% PFA for 1 h and then permeabilized in 0.2% Triton X-100 (Sigma-Aldrich, # 11332481001) for 30 min. Samples were then blocked in 2% BSA for 1 h and then incubated with primary antibody for 1 h. Samples were then labeled with donkey-raised secondary antibodies (1:400) for another 1 h. Cells were further analyzed using a Guava EasyCyte (Luminex) flow cytometer and its accompanied software.

### Statistical analysis

All experiments were conducted within = 3–5 replicates and repeated in $n > 3$ independent experiments. Statistical analysis was performed using Origin (https://www.Originlab.com). $p$ values were calculated using unpaired, two-sided Student's $t$ test. In all cases, a $p$ value of less than 0.05 was considered statistically significant.

### Microscopy

All confocal micrographs were acquired using an Olympus DSUIX81 spinning-disc confocal microscope equipped with an EMCCD camera (iXon X3, Andor) and the Zeiss LSM 880 confocal laser-scanning microscope and analyzed via Imaris x64 9.2.1.

## Reporting summary

Further information on research design is available in the Nature Portfolio Reporting Summary linked to this article.

## Data availability

Data supporting the findings of this study are available within the article and its Supplementary Information files and from the corresponding authors upon reasonable request. The scRNA-seq data generated in this study have been deposited in the Gene Expression Omnibus database under accession code GSE205611. Source data are provided with this paper.

## Code availability

MATLAB and RStudio scripts used in this work are available from corresponding authors upon request.

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

## Acknowledgements

This work is supported by the University of Michigan Mechanical Engineering Department, the Michigan-Cambridge Collaboration Initiative, the University of Michigan Mcubed Fund, the 21st Century Jobs Trust Fund received through the Michigan Strategic Fund from the State of Michigan (Grant CASE-315037), the National Science Foundation (I-Corps 2112458, CBET 1901718, and CMMI 2325361), and the National Institutes of Health (R21 HD100931, R21 NS113518, R21 HD105192, R21 HD109635, and R01HD079546). S.N.E. is partially supported by the National Institutes of Health T32 Graduate Research Fellowship under grant no. NIH T32 HD007505 and the University of Michigan Rackham Predoctoral Fellowship. A.M.R.I. is partially supported by the National Science Foundation Graduate Research Fellowship under grant no. DGE 1256260 and the University of Michigan

Rackham Predoctoral Fellowship. A.A. is supported by a Graduate Student Fellowship from the UCLA BSCRC. The derivation and differentiation of NOA hiPSC lines was supported by the Luca Bella Foundation to A.T.C. and S.S. This work was also supported by the Swedish Research Council (2016-01919) and Swedish Society for Medical Research (Dnr4-236-2107). S.P. holds the Canada Research Chair in Functional Genomics of Reproduction and Development (950-233204). We acknowledge the Michigan Advanced Genomics Core for scRNA-seq service. We are also appreciative of the MCDB/ BSCRC Imaging Core. We are grateful to Dr. Varsha Desai and Kaori Saito for conducting mycoplasma testing. In addition, we are grateful to Dr. Youngsun Hwang and Jonathan DiRusso for their guidance on optimizing screening protocol and confocal imaging.

## Author contributions

S.N.E. and J. Fu conceived and initiated the project; S.N.E. and Y.Z. designed, performed and quantified most experiments, including scRNA-seq data analysis and interpretation; A.A., T.J.H., N.A., J. Faith, M.J.L., M.S. and Q.Y.W. conducted experiments related to participant-derived samples and S.S. consented NOA participants; A.M.R.I. conducted live imaging and related analysis; X.X. and N.K. maintained cell culture, participated in experiments; Y.S. helped to design experiments; C.Z. helped with scRNA-seq data analysis under the supervision of S.P.; R.Y. and K.R.C. provided ISL1-KO cell line; A.T.C. supervised studies of NOA hiPSC lines; S.N.E., Y.Z., A.A., A.T.C. and J. Fu wrote the manuscript; J. Fu and A.T.C. supervised the entire project. All authors edited and approved the manuscript.

## Competing interests

A patent by J. Fu, Y.Z., Y.S. and S.N.E. related to this work has been granted (US11672832B2/WO2018106997). The derivation protocol of AMLCs from hPSCs is covered in this patent. Remaining authors declare no competing interests.
