## [Peer Review File · Nature Communications]

REVIEWER COMMENTS

Reviewer #1 (Remarks to the Author):

In this manuscript, Esfahani et al study the specification of human primordial germ cells (PGCs) in a 3D culture that mimics cellular interactions that take place in human embryos. In vitro protocols to generate human PGC-like cells have been generated over the past 7 years. The first protocols were based on the administration of exogenous growth factors to promote PGC specification, but recent ones have reported PGC specification as a consequence of inter-cellular interactions (Zheng et al, Nature, 2019) and geometrical constraints (Jo et al, eLife, 2022; Minn et al, eLife, 2020). Therefore, the main finding of the paper as highlighted in the title “Derivation of human PGC-like cells in an embryonic-like culture” is not novel. Moreover, Zheng et al already showed that amniotic epithelial cells promote a PGC fate in human pluripotent stem cells, and the involvement of the BMP, WNT, and NODAL pathways during human PGC-like cell specification is already known. The novel piece of information that this manuscript presents is the demonstration that the amnion factor ISL1, promotes PGC specification, potentially by controlling the expression of WNT and BMP ligands. Therefore, the impact is limited, and the paper would be more suitable for a specialised journal. There are also several issues with the current version of this manuscript:

1. The authors explore the roles of the BMP, WNT, and NODAL pathways during PGC-like cell specification in their in vitro culture system. Surprisingly, the WNT inhibitor IWP2 inhibited PGC specification, but the WNT inhibitor IWR1 had no effect. This result is not commented on or discussed in the paper. What is the conclusion of the authors? What explains the different phenotypes observed with the two inhibitors? Is the inhibitor IWR1 working? Can the findings be validated genetically?
2. In the co-culture experiments the inhibitor treatments do not allow to determine whether the effects are cell autonomous or non-cell autonomous. For example, is BMP inhibition preventing PGC-like cell specification directly, or by compromising amniotic epithelial cell fate specification? Can the ISL1 KO phenotype be rescued by upregulating WNT and/or BMP ligands in the amniotic epithelial cells? Are ISL1 KO ESCs able to form amniotic epithelial cells or their defect is specific to WNT/BMP expression?
3. The authors use their in vitro system to test whether patients suffering from azoospermia can form PGC-like cells, and observe no differences compared to controls. Given that validating the findings in vivo is impossible, the observation remains speculative. Does the 3D in vitro culture system described by the authors provide any advantage in terms of screening for disease phenotypes? Couldn't the same experiment be done in the classical embryoid-body based differentiation protocol? Are they proposing that this method could be used to form PGC-like cells from otherwise infertile patients?
4. How does amnion formation in this system compare to other protocols of amnion specification? For example, do amniotic epithelial cells acquire a more mature signature and express markers such as GABRP?

5. Figure 3c (day 4): it seems that PGC-like cells are located far away from the amniotic epithelial cells. Is this the case? Do amniotic epithelial cells change their identity/character when they are exposed to mTESR in co-culture conditions? Is there a minimum number of amniotic epithelial cells required to induce a PGC-like fate?

6. Detailed quantifications/analyses are missing from most of the immunofluorescence images. For example, the data on the patient-derived PGC-like cells is not analysed at all.

Minor comments:

1. All catalogue numbers are missing in the methods section

2. The rationale to classify structures into irregular, columnar, and squamous is not clear. Do PGCs only appear in irregular structures? In Figure 1c it would be useful to indicate the % of structures that are squamous or columnar, not just irregular.

3. The finding that the percentage of Geltrex added to the culture has such a profound influence on PGC specification efficiency (Fig. 1c) is quite surprising. Could the authors provide some potential explanation?

4. Extended Data Figure 8: this shows one of the very few analyses of IF data. The methods section indicates that IF images were analysed with Fiji. How?

5. Extended Data Figure 9: it is not clear how the cells were tracked across time (if at all). How was the information on cell lineages superimposed on the IF data so that the same cells were found? Could the authors provide some videos?

Reviewer #2 (Remarks to the Author):

The origin of the Primordial Germ Cells (PGCs) in human is somewhat mired in controversy. The lack of direct experimental data leads to a reliance on the interpretation of pictures of embryos from the Carnegie collection. However, this is fraught with difficult interpretations. Over the last few years a number of embryonic stem cell derived in vitro systems have been put together that reflect early human development but, so far, these systems have not been helpful as the PGCs arise from the boundary between the amnion and the epiblast. Being very motile, these cells move away from their point of origin and can be easily found spread around the epiblast and the amnion. Cases have been made for both tissues as their origin. In this manuscript Esfahani and colleagues revisit this question from the perspective of a serendipitous finding that they leverage as a method to reveal their origin.

The authors create an in vitro culture of hESCs that generates amnion and epiblast under conditions that produce PGCs. Using the system they show rather convincingly that PGCs arise from the epiblast under

the influence of the amnion. The results are convincing and important. The chemical experiments are, for the most part clear, particularly those associated with BMP. Those related to Wnt signalling should take into account the effect that Tankyrase inhibitors have on YAP signalling. For this reason and the authors should comment on the differential effects that IWR1 and IWP2 have on the PGCs. Could YAP be involved in this process and, if so, how?

Reviewer #3 (Remarks to the Author):

Esfani et al. reported derivation of hPGCLCs in a 3D culture system they termed Gel-3D. They suggested that amniotic ectoderm-like cells (AMLCs) are derived in the system and the AMLCs further induced hPGCLCs. Although it is logical to suggest that the small populations of AMLCs might induce the first population of hPGCLCs, the evidences and datasets supported this notion are weak. The authors showed 4 main figures of dataset with very limited experimental results to support their main conclusion. Overall, the study is quite preliminary, many results are ambiguous without proper controls.

Major concerns:

1. If hPGCLCs were induced by BMP4 secreted by AMLCs, why would the authors added BMP4 in the first stage and their derivation (Fig3C) and removed them in their later derivation? The induction of PGCLCs might depend on the initial induction of BMPs, but not on the addition of AMLCs in the later culture, or the AMLCs might only partially promote the induction of PGCLCs.
2. Where are the controls in Figure 3C, and d? Are they counted as one set of experiment? Why using two different set of time points in these two figures?
3. The authors stated 'Immunofluorescence analysis reveals drastic reductions of hPGCLC numbers on both day 3 and 155 day 8 in the Gel-3D culture generated from an ISL1-knockout (KO) hPSC line (Fig. 3d).' It is not clear how the authors reached the conclusion of 'drastic reduction' with what measurement.
4. It is not clear what is the advantage and purpose of using Gel3D system to screen the NOA patient sample. Is the author trying to screen for someone who has abnormal amniotic ectoderm development that might affect PGCLC number? Otherwise, it is not surprise that all of the NOA patient has similar PGCLCs.

Reviewer #4 (Remarks to the Author):

Review reports for "Derivation of Human Primordial Germ Cell-Like Cells in an Embryonic-Like Culture" by Esfahani et al. (NCOMMS-23-03065-T)

In this study, the authors improved a 3D ECM culture system (Gel-3D culture) of amnion-like cells (AMLC) to facilitate induction of human primordial germ-like cells (hPGCLCs). By using this improved 3D culture system, hPGCLCs were induced in close proximity to AMLCs, which is suggested to be morphologically similar to the in vivo PGC specification process. Through scRNA-seq analysis, the authors showed that hPGCLCs derived from the new 3D culture system have similar gene expression profiles to those of hPGCLCs generated using existing methods, and these profiles correlate with in vivo hPGCs to a similar degree as existing protocols. The authors also demonstrate that the expression of ISL in AMLC is required for hPGCLC induction mediated by BMP4, and inhibitor experiments reveal the requirement of WNT signaling and the intermediation of ALK2/3 in BMP signaling during hPGCLC induction. Moreover, the authors reveal the dynamics of cellular lineage during hPGCLC induction, and they also successfully generated iPSC from NOS patients and induced hPGCLCs using the new method. Overall, this study effectively mimics the in vivo specification of hPGC, taking into account the developmental context. I feel this system interesting. Here, I would like to try listing a few concerns about this manuscript.

- The authors reported that the efficiency of hPGCLC induction and gene expression profile of the induced hPGCLCs were similar to those generated using existing methods. This indicates that while the Gel-3D culture improved the AMLC induction method so as to induce PGCLCs, it did not necessarily improve the efficiency of hPGCLC induction. While the morphological correlation of hPGCLC induction to in vivo PGC specification is interesting, the authors may need to further clarify how this new method can serve as an improved tool for understanding PGC specification, in addition to its compatibility with live imaging. A possible approach could be to compare the efficiency of drug perturbation between their method and conventional aggregation culture, which might help in identifying any advantages or limitations of their new method.

- The authors developed a co-culture system of hPSCs and AMLCs to investigate the requirement of paracrine signaling from AMLCs to PSCs. While this method is novel, the AMLC induction protocol employed was different from that used in the Gel-3D culture system. Could the authors clarify how the AMLCs induced using this protocol are similar to in vivo amnion or AMLCs induced using other protocols, such as in terms of gene expression profiles.?

- In addition, I think it is already known that BMP4, NODAL, and WNT signaling play important roles in PGC/PGCLC induction in humans and/or mice. Moreover, previous studies (e.g., Sasaki et al., 2016 Dev Cell) have demonstrated that the nascent amnion is the source of BMP4. Therefore, could the authors consider clarification on how the new method improves our understanding of the signaling mechanisms involved in hPGCLC induction?

- The authors reported successful induction of hPGCLCs from NOS iPSCs using their new method. However, it would be valuable to understand how this new method has improved the induction of hPGCLCs from NOS iPSCs, specifically in terms of induction efficiency and quality. In this regard, could the authors clarify how the induction efficiency and/or quality of hPGCLCs was improved by their new method compared to a conventional method?

Response to Reviewer 1

General comment (1): *In this manuscript, Esfahani et al study the specification of human primordial germ cells (PGCs) in a 3D culture that mimics cellular interactions that take place in human embryos. In vitro protocols to generate human PGC-like cells have been generated over the past 7 years. The first protocols were based on the administration of exogenous growth factors to promote PGC specification, but recent ones have reported PGC specification as a consequence of inter-cellular interactions (Zheng et al, Nature, 2019) and geometrical constraints (Jo et al, eLife, 2022; Minn et al, eLife, 2020). Therefore, the main finding of the paper as highlighted in the title “Derivation of human PGC-like cells in an embryonic-like culture” is not novel. Moreover, Zheng et al already showed that amniotic epithelial cells promote a PGC fate in human pluripotent stem cells, and the involvement of the BMP, WNT, and NODAL pathways during human PGC-like cell specification is already known. The novel piece of information that this manuscript presents is the demonstration that the amnion factor ISL1, promotes PGC specification, potentially by controlling the expression of WNT and BMP ligands. Therefore, the impact is limited, and the paper would be more suitable for a specialised journal. There are also several issues with the current version of this manuscript:*

Response: We appreciate the reviewer for his / her concise summary of this study. Having said that, we respectfully disagree with the reviewer on the novelty / impact of this work. Conventional protocols utilizing exogenous growth factors for hPGCLC induction require the conversion of primed hPSCs into either a naïve-like or mesoderm-like state and the generation of embryoid bodies in low-adhesion, pyramid-shaped wells. These protocols are not very convenient to implement and are incompatible with live imaging and thus imaging-based screening pipelines. Furthermore, the *in vivo* relevance of these protocols remains doubtful. As mentioned by the reviewer, in recent publications hPGCLCs have been derived using either microfluidic devices (Zheng et al. Nature, 2019) or micropatterned adhesive surfaces (Jo et al. eLife, 2022; Minn et al. eLife, 2020). However, these new methods for hPGCLC induction remain challenging to implement in typical biological labs. In comparison, the Gel-3D culture reported in this work significantly simplifies hPGCLC induction protocols and is very convenient for quantification and imaging. Furthermore, it can be easily implemented in typical biological labs. As shown in the manuscript, the Gel-3D culture can be easily implemented in multiwell plate formats and as such is compatible with screening applications. Given these advantages of the Gel-3D culture, we strongly believe that it has the great potential to be widely adopted by the research community for hPGCLC induction and related mechanistic and translational studies.

We should also note that the three works mentioned by the reviewer (Zheng et al. Nature, 2019; Jo et al. eLife, 2022; Minn et al. eLife, 2020) only report the observation of co-development of AMLCs and hPGCLCs in their culture systems but did not draw conclusions or provide concrete evidence on the inductive role of AMLCs in hPGCLC specification. In this work, besides reporting a new, easily implementable hPGCLC induction protocol, we have further, for the first time, obtained data supporting that AMLCs derived from hPSCs could induce hPGCLC specification through paracrine signaling downstream of *ISL1*. This important new knowledge is further strengthened by additional new experiments conducted for this resubmission. The new

results and related discussions have been added to the manuscript (Fig. 3h&i and Extended Data Fig. 7e).

We agree with the reviewer that the involvements of BMP, WNT, and NODAL signaling in PGC/PGCLC specification have been established in the field. In this work, the investigations of the roles of these pathways are to understand their functional connections with *ISL1*. As such, related data have only been included in Extended Data Figures. To clarify these issues, we have revised the manuscript accordingly (Lines 168-182, 185-193, 197-198, 228-229, and 245-256 on pages 9-13).

Specific comment (1): *The authors explore the roles of the BMP, WNT, and NODAL pathways during PGC-like cell specification in their in vitro culture system. Surprisingly, the WNT inhibitor IWP2 inhibited PGC specification, but the WNT inhibitor IWR1 had no effect. This result is not commented on or discussed in the paper. What is the conclusion of the authors? What explains the different phenotypes observed with the two inhibitors? Is the inhibitor IWR1 working? Can the findings be validated genetically?*

Response: We thank the reviewer for pointing out this discrepancy. In the first submission, we used IWR1 at a concentration of 10 μ M, a widely used dose in WNT-related research, and did not observe any inhibitory effect on hPGCLC specification. Per the reviewer's comments, we have conducted new experiments using IWR1 at a higher concentration of 30 μ M. In both Gel-3D cultures and AMLC-hPSC co-cultures, hPGCLC specification is inhibited by 30 μ M IWR1. To corroborate this finding, we tested another WNT inhibitor, XAV939, which targets β -CATENIN by inhibiting tankyrase. hPGCLC specification in both Gel-3D cultures and AMLC-hPSC co-cultures is inhibited by XAV939. Together, our data support the role of WNT signaling in hPGCLC specification. In this resubmission, new data have been included in Extended Data Fig. 9&10. We have also revised the main text accordingly (Lines 185-193 on page 10).

Specific comment (2): *In the co-culture experiments the inhibitor treatments do not allow to determine whether the effects are cell autonomous or non-cell autonomous. For example, is BMP inhibition preventing PGC-like cell specification directly, or by compromising amniotic epithelial cell fate specification?*

Response: We appreciate the reviewer for this comment and apologize for the confusion. In AMLC-hPSC co-culture studies, drug inhibitors were only introduced after AMLC differentiation (48 h of BMP4 treatment on hPSCs). Thus, the introduction of inhibitors should only interfere paracrine interactions between AMLCs and hPSCs. To clarify whether the drug inhibitors might alter AMLC identity, we conducted additional experiments by treating AMLCs alone with the inhibitors for 48 h, before staining the cells for amnion markers, including TFAP2A, GATA3, and ISL1. Based on squamous cell morphology and positive immunostaining results, we conclude that the drug inhibitors did not alter the identity of AMLCs. These new data have now been included in Extended Data Fig. 10. We have further revised the main text to clarify the drug inhibition assays for AMLC-hPSC co-cultures (Lines 191-193 on page 10).

Can the ISL1 KO phenotype be rescued by upregulating WNT and/or BMP ligands in the amniotic epithelial cells?

We are grateful to the reviewer for this insightful question. To address this comment, we conducted new rescue assays by adding BMP4, WNT3A, or both morphogens to Gel-3D cultures generated from *ISLI*-KO hPSCs. We tested a range of concentrations for BMP4 (25 - 100 ng/mL) and WNT3A (50 - 500 ng/mL). Our data show that supplementing 25 ng/mL of BMP4 effectively rescues hPGCLC specification in Gel-3D cultures of *ISLI*-KO hPSCs. However, higher concentrations of BMP4 (> 50 ng/mL) drive all *ISLI*-KO hPSCs into AMLCs in the Gel-3D culture, and therefore no PGCLCs were observed. For WNT3A, concentrations greater than 250 ng/mL would rescue hPGCLC specification in Gel-3D cultures of *ISLI*-KO hPSCs. In Gel-3D cultures of *ISLI*-KO hPSCs where both BMP4 and WNT3A were added, we observed similar extents of hPGCLC specification rescuing, as compared to adding BMP4 or WNT3A alone.

We further validated rescuing effects of BMP4 and WNT3A on hPGCLC specification using co-cultures of *ISLI*-KO AMLCs and wide-type hPSC clusters. Consistent with the data from Gel-3D cultures, hPGCLC specification was rescued in the co-culture assays upon addition of either 25 ng/mL BMP4, 250 ng/mL WNT3A, or a combination of 50 ng/mL BMP4 and 250 ng/mL WNT3A. These new data have been included in the updated manuscript (Fig. 3h&i and Extended Data Fig. 7e). We have also added related discussions to the main text (Lines 177-182, and 400-404 on pages 9 & 20).

Are ISLI KO ESCs able to form amniotic epithelial cells or their defect is specific to WNT/BMP expression?

We appreciate the reviewer's comment. To address this question, *ISLI*-KO hPSCs were used in both Gel-3D cultures and 2D AMLC differentiation assays (by treating *ISLI*-KO hPSCs with BMP4 for 48 h). Immunostaining for amnion markers TFAP2A/B and GATA3 confirmed that *ISLI*-KO hPSCs retain the capacity to give rise to AMLCs in both Gel-3D cultures and 2D AMLC differentiation assays. These new data have now been included in Extended Data Fig. 7b&c.

Additionally, we performed RT-qPCR to examine expression levels of *BMP2/4*, *WNT5A/5B/6*, *TGF- β* , and *NODAL* in AMLCs generated in 2D cultures using *ISLI*-KO hPSCs and wide-type controls. As shown in Fig. 3g, expression levels of *BMP2/4* and *WNT5A/5B/6* are decreased in *ISLI*-KO AMLCs as compared to wide-type AMLCs. Thus, our data show that *ISLI*-KO hPSCs can still give rise to AMLCs, even though *ISLI*-KO AMLCs show reduced RNA expression of WNT and BMP ligands, consistent with the reduced potency of *ISLI*-KO AMLCs for inducing hPGCLCs from hPSCs. We have added these discussions to the main text (Lines 174-177 on page 9).

Specific comment (3): *The authors use their in vitro system to test whether patients suffering from azoospermia can form PGC-like cells, and observe no differences compared to controls. Given that validating the findings in vivo is impossible, the observation remains speculative.*

Response: We appreciate the reviewer's comment. We included data from azoospermia patients in our study to show the utility of our Gel-3D system for screening applications. We totally agree

with the reviewer that our data only show that hiPSCs derived from azoospermia patients can be used to derive hPGCLCs. This conclusion was already included in our first submission.

Does the 3D in vitro culture system described by the authors provide any advantage in terms of screening for disease phenotypes? Couldn't the same experiment be done in the classical embryoid-body based differentiation protocol?

We appreciate the reviewer's question. In principle, existing embryoid body-based differentiation protocols for obtained hPGCLCs can be used for screening for disease phenotypes. However, as we mentioned earlier, conventional protocols for generating hPGCLCs require embryoid body-like cultures in low-adhesion pyramid-shaped wells, which are not very convenient to implement and are incompatible with live imaging or imaging-based screening pipelines. In comparison, the Gel-3D culture reported in this work significantly simplifies hPGCLC induction protocols and is very convenient for quantification and imaging. Furthermore, it can be easily implemented in typical biological labs. As shown in the manuscript, the Gel-3D culture can be easily implemented in multiwell plate formats and as such is compatible with imaging-based screening applications. Given these advantages of the Gel-3D culture, we strongly believe that it has the great potential to be widely adopted by the research community for hPGCLC induction and related mechanistic and translational studies.

Are they proposing that this method could be used to form PGC-like cells from otherwise infertile patients?

We thank the reviewer for this comment. Our data only support that the Gel-3D culture could be used to obtain hPGCLCs from azoospermia patient-derived hiPSCs. To avoid any confusion, we have stressed this point in the manuscript (Lines 228-229 on page 11).

Specific comment (4): How does amnion formation in this system compare to other protocols of amnion specification? For example, do amniotic epithelial cells acquire a more mature signature and express markers such as GABRP?

Response: We appreciate the reviewer for raising this important point. Based on our comparative transcriptome analysis in Extend Data Fig. 5b, the transcriptome of AMLCs generated in Gel-3D cultures correlates very well with AMLCs generated in Zheng *et al.* 2019¹. Regarding expression of mature amnion markers, the transcriptome of day 3 AMLCs generated in Gel-3D cultures does show *GABRP* and *IGFBP3* expression. These data are included in Extended Data Fig. 5b. To clarify this point, we have added these discussions in the manuscript (Lines 117-119, and 138-141 on pages 7 & 8).

Specific comment (5): Figure 3c (day 4): it seems that PGC-like cells are located far away from the amniotic epithelial cells. Is this the case?

Response: We apologize for the confusion. In AMLC-hPSC co-cultures, specification of hPGCLCs from hPSCs is triggered by paracrine signals originated from AMLCs. Since hPSC clusters are seeded after AMLC differentiation, the experimenters have no control over the

distance between later added hPSC clusters and AMLCs. Thus, some PGCLCs might appear far away from AMLCs.

Do amniotic epithelial cells change their identity/character when they are exposed to mTESR in co-culture conditions?

We appreciate the reviewer's question. To exclude this possibility, we conducted new assays in which AMLCs were first derived from hPSCs by treating the cells with BMP4 for 48 h. The resulting AMLCs were then cultured in mTeSR for an additional 8 days before immunostaining for amnion markers, including TFAP2A/B, ISL1, and GATA3. Our immunostaining data show that after 8 days of culture in mTeSR, AMLCs continue to proliferate and remain positive for the amnion markers, supporting that these cells remain the AMLC identity. These new data have now been included in Extended Data Fig. 6a, with corresponding discussions added to the main text (Lines 155-157 on page 8).

Is there a minimum number of amniotic epithelial cells required to induce a PGC-like fate?

Response: We appreciate the reviewer for raising this important question. To address this comment, we performed additional AMLC-hPSC co-culture experiments by varying the number of AMLCs in the co-culture. Our data show that the efficiency of hPGCLC induction within hPSC clusters increases with the number of AMLCs present in the co-culture. These new data have been included in Extended Data Fig. 6c of the revised manuscript, with related discussions added to the main text (Lines 162-163 on page 9).

Specific comment (6): Detailed quantifications/analyses are missing from most of the immunofluorescence images. For example, the data on the patient-derived PGC-like cells is not analysed at all.

Response: We thank the reviewer for this comment. To address this comment, we have added Fig. 4c, which presents the percentage of hPGCLCs generated from azoospermia patient-derived hiPSCs using the Gel-3D culture. In addition, we have quantified data generated from *ISL1*-KO hPSCs in Fig. 3e, with related discussions added to the main text (Lines 168-170, and 238 on pages 9 & 12).

Minor comments:

Specific comment (7): All catalogue numbers are missing in the methods section.

Response: We thank the reviewer for raising this concern. Accordingly, we have added all catalog numbers in the Methods section of the manuscript.

Specific comment (8): The rationale to classify structures into irregular, columnar, and squamous is not clear. Do PGCs only appear in irregular structures?

Response: We thank the reviewer for this comment. In the Gel-3D culture, columnar epiblast-like cysts are characterized by a thick epithelium comprised of tall elongated epithelial cells (undifferentiated hPSCs). Squamous cysts exhibit a squamous epithelial morphology with

flattened cells and a reduced epithelium thickness, comprised of AMLCs. Irregular cysts are defined as cysts whose morphologies differ from those of squamous and columnar cysts. The vast majority of hPGCLCs emerge in irregular cysts. To better clarify these different structures in the Gel-3D culture, we have added some new discussions to the main text (Lines 73-75 on page 5).

In Figure 1c it would be useful to indicate the % of structures that are squamous or columnar, not just irregular.

We thank the reviewer for raising this point. Accordingly, the percentages of columnar epiblast-like cysts and squamous amniotic ectoderm-like cysts have now been added to Fig. 1c.

Specific comment (9): *The finding that the percentage of Geltrex added to the culture has such a profound influence on PGC specification efficiency (Fig. 1c) is quite surprising. Could the authors provide some potential explanation?*

Response: We sincerely appreciate the reviewer for this comment. We are also surprised by this discovery. Our previous studies show that 3D Geltrex overlay added to 2D hPSC cultures can effectively trigger differentiation of AMLCs from hPSCs^{2,3}. The data presented in the current work suggest that the timing of adding Geltrex overlay and its concentration are important factors that likely establish an amniogenic environment, but this amniogenic environment remains conducive for hPSCs to choose a different lineage development path towards hPGCLCs, under paracrine inductive effects from nascent AMLCs. We have added this discussion to the revised manuscript (Lines 73-75 on page 5).

Specific comment (10): *Extended Data Figure 8: this shows one of the very few analyses of IF data. The methods section indicates that IF images were analysed with Fiji. How?*

Response: We thank the reviewer for this comment. All immunofluorescent images were taken using the same confocal microscope with the same setup. The images were then imported into Fiji software and processed using identical parameters. The number of TFAP2C+NANOG+SOX17+ cells was manually counted as hPGCLCs. DAPI staining was utilized to determine the total cell count. We calculated the percentage of hPGCLCs in each image. To avoid confusion, we revised the Method section ‘Quantification of immunofluorescent images’ (Lines 383-389 on page 19).

Specific comment (11): *Extended Data Figure 9: it is not clear how the cells were tracked across time (if at all). How was the information on cell lineages superimposed on the IF data so that the same cells were found? Could the authors provide some videos?*

Response: We thank the reviewer for this comment and recognize that our explanation of the lineage tracing required more clarity. As explained in the Methods section ‘Live cell video analysis’, the live cell video analysis was done using a method developed by Resto-Irizarry et al⁴. In brief, live cell videos were analyzed with a Python pipeline that carries out pre-processing, identification of individual cell nuclei, and tracking from one time point to another using Euclidean distance. The pipeline also identifies cell division events and stores parent and

daughter cell IDs to carry out lineage tracing. The final video frame is compared to immunostained samples to ascertain cell identity (hPGCLCs vs. non-hPGCLCs). The Python pipeline lineage information is then used to create the network schematic shown in Extended Data Fig. 11a, where each plane is a network representation of an imaged cyst at a specific time point. Cells in the hPGCLC lineage are highlighted with a yellow color, and solid black lines indicate a direct lineage connection with the final hPGCLC being traced. More detail has been added to the main text and Method section ‘Live cell video analysis’ to improve the clarity (Lines 213-216, and 536-540 on pages 11 & 26). We also provided a video that can better show cell tracking.

Response to Reviewer 2

General comment (1): *The origin of the Primordial Germ Cells (PGCs) in human is somewhat mired in controversy. The lack of direct experimental data leads to a reliance on the interpretation of pictures of embryos from the Carnegie collection. However, this is fraught with difficult interpretations. Over the last few years a number of embryonic stem cell derived in vitro systems have been put together that reflect early human development but, so far, these systems have not been helpful as the PGCs arise from the boundary between the amnion and the epiblast. Being very motile, these cells move away from their point of origin and can be easily found spread around the epiblast and the amnion. Cases have been made for both tissues as their origin. In this manuscript Esfahani and colleagues revisit this question from the perspective of a serendipitous finding that they leverage as a method to reveal their origin.*

The authors create an in vitro culture of hESCs that generates amnion and epiblast under conditions that produce PGCs. Using the system they show rather convincingly that PGCs arise from the epiblast under the influence of the amnion. The results are convincing and important.

Response: We thank the reviewer for the positive summary of this work and his / her recognition of the significance of our study.

Specific comment (1): *The chemical experiments are, for the most part clear, particularly those associated with BMP. Those related to Wnt signalling should take into account the effect that Tankyrase inhibitors have on YAP signalling. The authors should comment on the differential effects that IWR1 and IWP2 have on the PGCs. Could YAP be involved in this process and, if so, how?*

Response: We appreciate the reviewer's insightful comment. While the Hippo/YAP signaling is a potent regulator of cell differentiation in various developmental contexts⁵⁻⁸, its role in PGC induction is not well understood. Herein, to examine potential involvement of YAP signaling in the Gel-3D culture, we performed immunostaining for YAP1 on day 3. Our data do not show YAP nuclear translocation in any of the cells in the Gel-3D culture (Figure 1a). To further explore the involvement of YAP signaling, we supplemented two different YAP inhibitors at varying concentrations in the Gel-3D culture. Cerivastatin, known to inhibit YAP through the mevalonate pathway, was used at concentrations ranging from 0.1 - 1 μ M. Dobutamine, which inhibits YAP1 through β -adrenergic receptor, was used at concentrations ranging from 1 - 10 μ M. Our data show that the two inhibitors did not affect cell differentiation in the Gel-3D culture (Figure 1b&c). To further support these findings, we utilized the microfluidic amniotic sac embryoid (PASE) system generated by Zheng *et al.* 2019¹, which recapitulates successive key early human post-implantation developmental landmarks including amnion and PGC specification. Treatment with Cerivastatin or Dobutamine did not result in significant differences in hPGCLC specification in the microfluidic PASE system (Figure 1d). Together, our data suggest that it is unlikely that YAP signaling plays a role in the Gel-3D culture.

We thank the reviewer for pointing out the discrepancy between IWR1 and IWP2 treatments. In the first submission, we used IWR1 at a concentration of 10 μ M, a widely used dose in WNT-related research, and did not observe any inhibitory effect on hPGCLC specification. Per the

reviewer's comments, we have conducted new experiments using IWR1 at a higher concentration of 30 μM . In both Gel-3D cultures and AMLC-hPSC co-cultures, hPGCLC specification is inhibited by 30 μM IWR1. To corroborate this finding, we tested another WNT inhibitor, XAV939, which targets β -CATENIN by inhibiting tankyrase. hPGCLC specification in both Gel-3D cultures and AMLC-hPSC co-cultures is inhibited by XAV939. Together, our data support the role of WNT signaling in hPGCLC specification. In this resubmission, new data have been included in Extended Data Fig. 9&10. We have also revised the main text accordingly (Lines 185-193 on page 10).

Figure 1. hPGCLC induction in the Gel-3D culture is not inhibited by YAP inhibitors. **a.** Representative micrographs showing immunostaining for YAP1 and SOX17 for the Gel-3D culture on day 3. **b.** Micrographs showing immunostaining for TFAP2C, NANOG, and SOX17; and TFAP2A, GATA3, and SOX17 for the Gel-3D culture with Cerivastatin or Dobutamine treatment as indicated. **c.** Plot showing percentages of TFAP2C+SOX17+ hPGCLCs in the Gel-3D culture on day 3 as a function of Cerivastatin or Dobutamine treatments as indicated. **d.** Micrographs showing immunostaining for TFAP2C, NANOG, and SOX17 for the microfluidic PASE model at $t = 48\text{h}$ with Cerivastatin or Dobutamine treatment as indicated.

Response to Reviewer 3

General comment (1): *Esfahani et al. reported derivation of hPGCLCs in an 3D culture system they termed Gel-3D. They suggested that amniotic ectoderm-like cells (AMLCs) are derived in the system and the AMLCs further induced hPGCLCs. Although it is logical to suggest that the small populations of AMLCs might induce the first population of hPGCLCs, the evidences and datasets supported this notion are weak. The authors showed 4 main figures of dataset with very limited experimental results to support their main conclusion. Overall, the study is quite preliminary, many results are ambiguous without proper controls.*

Response: We appreciate the reviewer's comments. Here we like to reiterate the technical advantages of our Gel-3D culture for hPGCLC induction and the scientific insights included in this study. In this resubmission, we have included additional experiments to strengthen our conclusions and provide more explanations to clarify related results.

Conventional protocols utilizing exogenous growth factors for hPGCLC induction require the conversion of primed hPSCs into either a naïve-like or mesoderm-like state and the generation of embryoid bodies in low-adhesion, pyramid-shaped wells. These protocols are not very convenient to implement and are incompatible with live imaging and thus imaging-based screening pipelines. Furthermore, the *in vivo* relevance of these protocols remains doubtful. As mentioned by the reviewer, in recent publications hPGCLCs have been derived using either microfluidic devices (Zheng *et al. Nature*, 2019) or micropatterned adhesive surfaces (Jo *et al. eLife*, 2022; Minn *et al. eLife*, 2020). However, these new methods for hPGCLC induction remain challenging to implement in typical biological labs. In comparison, the Gel-3D culture reported in this work significantly simplifies hPGCLC induction protocols and is very convenient for quantification and imaging. Furthermore, it can be easily implemented in typical biological labs. As shown in the manuscript, the Gel-3D culture can be easily implemented in multiwell plate formats and as such is compatible with screening applications. Given these advantages of the Gel-3D culture, we strongly believe that it has the great potential to be widely adopted by the research community for hPGCLC induction and related mechanistic and translational studies.

Besides establishing the Gel-3D culture, this study further present mechanistic data to show that AMLCs in the Gel-3D culture drive specification of hPGCLCs in the culture through paracrine induction downstream of *ISL1*, likely involving BMP and WNT signaling. Using genetic and drug perturbation assays, we further show the roles of NODAL, WNT, and BMP pathways in hPGCLC specification in the Gel-3D culture. As a proof of demonstration for the utility of the Gel-3D culture for screening applications, we have successfully implemented the Gel-3D culture in multiwell plate formats for screening of hPGCLC induction from eight azoospermia patient-derived hiPSC lines.

As mentioned, in this resubmission we have included additional experiments to strengthen our conclusions and provide more explanations to clarify related results. Hence, this work not only presents a new useful method for hPGCLC induction from hPSCs, but also contribute to the current understanding of hPGCLC development, particularly in the context of human development.

Major concerns:

Specific comment (1): *If hPGCLCs were induced by BMP4 secreted by AMLCs, why would the authors add BMP4 in the first stage and their derivation (Fig3C) and removed them in their later derivation? The induction of PGCLCs might depend on the initial induction of BMPs, but not on the addition of AMLCs in the later culture, or the AMLCs might only partially promote the induction of PGCLCs.*

Response: We appreciate the reviewer's comment and apologize for any confusion. In the AMLC-hPSC co-culture studies (Fig. 3c), hPSCs are first treated with BMP4 for 48 h to obtain AMLCs. Subsequently, BMP4 is removed before clusters of undifferentiated hPSCs are introduced to AMLCs to establish their co-cultures. During AMLC-hPSC co-cultures, exogenous BMP4 need to be removed, since otherwise all hPSCs will differentiate into AMLCs without hPGCLC induction. We apologize for any initial lack of clarification regarding this experimental design. In the revised manuscript, we have provided further clarification in the main text to avoid confusion (Lines 154-159 on page 8).

Specific comment (2): *Where are the controls in Figure 3C, and d? Are they counted as one set of experiment? Why using two different set of time points in these two figures?*

Response: We appreciate the reviewer's comment. For these two figure panels, we had incorporated appropriate controls. In Fig. 3c, we included a control, in which hPSCs were cultured in mTeSR without the presence of AMLCs. In Fig. 3d, we included wild-type hPSCs as a control to establish Gel-3D cultures. We have revised related statements in the main text accordingly (Lines 161-162, and 169-170 on pages 8 & 9).

Regarding different time points, we apologize for the confusion here. Fig. 3c and Fig. 3d represent two distinct experiments. Fig. 3c shows the results of AMLC-hPSC co-culture studies to demonstrate the potential of AMLCs to induce hPGCLC induction within hPSC clusters. Fig. 3d shows the results of using *ISLI*-KO hPSCs in Gel-3D cultures, highlighting the role of *ISLI* in hPGCLC specification. In Fig. 3c, the day 2 data illustrate the presence of AMLCs in the 2D culture following a 48-h treatment with BMP4. The day 4 data show results after AMLC-hPSC co-cultures for 48 h. In Fig. 3d, *ISLI*-KO hPSCs were used in the Gel-3D culture, and we examined cell lineage differentiation on day 3 and day 8, the same time points that have been used in all Gel-3D studies. We have now clarified these points in the main text of the manuscript (Lines 154-157, and 169 on pages 8 & 9).

Specific comment (3): *The authors stated 'Immunofluorescence analysis reveals drastic reductions of hPGCLC numbers on both day 3 and 155 day 8 in the Gel-3D culture generated from an ISLI-knockout (KO) hPSC line (Fig. 3d).' It is not clear how the authors reached the conclusion of 'drastic reduction' with what measurement.*

Response: We appreciate the reviewer for raising this comment. In response to the reviewer's question, we quantified the percentage of hPGCLCs in the Gel-3D culture generated from both *ISLI*-KO and wild-type hPSCs, with the data plotted in Fig. 3e in the revised manuscript. We have also revised the main text accordingly (Lines 168-170 on page 9).

Specific comment (4): It is not clear what is the advantage and purpose of using Gel3D system to screen the NOA patient sample. Is the author trying to screen for someone who has abnormal amniotic ectoderm development that might affect PGCLC number? Otherwise, it is not surprise that all of the NOA patient has similar PGCLCs.

Response: We thank the reviewer for this comment. We included data from azoospermia patients in our study to show the utility of our Gel-3D system, which can readily handle multiple cell lines simultaneously for quantitative screening for hPGCLC differentiation. The results only suggest that hiPSCs derived from azoospermia patients can be used to generate hPGCLCs.

Response to Reviewer 4

General comment (1): *In this study, the authors improved a 3D ECM culture system (Gel-3D culture) of amnion-like cells (AMLC) to facilitate induction of human primordial germ-like cells (hPGCLCs). By using this improved 3D culture system, hPGCLCs were induced in close proximity to AMLCs, which is suggested to be morphologically similar to the in vivo PGC specification process. Through scRNA-seq analysis, the authors showed that hPGCLCs derived from the new 3D culture system have similar gene expression profiles to those of hPGCLCs generated using existing methods, and these profiles correlate with in vivo hPGCs to a similar degree as existing protocols. The authors also demonstrate that the expression of ISL in AMLC is required for hPGCLC induction mediated by BMP4, and inhibitor experiments reveal the requirement of WNT signaling and the intermediation of ALK2/3 in BMP signaling during hPGCLC induction. Moreover, the authors reveal the dynamics of cellular lineage during hPGCLC induction, and they also successfully generated iPSC from NOS patients and induced hPGCLCs using the new method. Overall, this study effectively mimics the in vivo specification of hPGC, taking into account the developmental context. I feel this system interesting. Here, I would like to try listing a few concerns about this manuscript.*

Response: We thank the reviewer for his / her concise summary of this work and his / her recognition of the significance of our study.

Specific comment (1): *The authors reported that the efficiency of hPGCLC induction and gene expression profile of the induced hPGCLCs were similar to those generated using existing methods. This indicates that while the Gel-3D culture improved the AMLC induction method so as to induce PGCLCs, it did not necessarily improve the efficiency of hPGCLC induction. While the morphological correlation of hPGCLC induction to in vivo PGC specification is interesting, the authors may need to further clarify how this new method can serve as an improved tool for understanding PGC specification, in addition to its compatibility with live imaging. A possible approach could be to compare the efficiency of drug perturbation between their method and conventional aggregation culture, which might help in identifying any advantages or limitations of their new method.*

Response: We thank the reviewer for this comment. Indeed, the Gel-3D culture reported in this work for hPGCLC induction represents a substantial improvement over previous hPGCLC induction systems. Conventional protocols utilizing exogenous growth factors for hPGCLC induction require the conversion of primed hPSCs into either a naïve-like or mesoderm-like state and the generation of embryoid bodies in low-adhesion, pyramid-shaped wells. These protocols are not very convenient to implement and are incompatible with live imaging and thus imaging-based screening pipelines. Furthermore, the *in vivo* relevance of these protocols remains doubtful. In comparison, the Gel-3D culture reported in this work significantly simplifies hPGCLC induction protocols and is very convenient for quantification and imaging. Furthermore, it can be easily implemented in typical biological labs. As shown in the manuscript, the Gel-3D culture can be easily implemented in multiwell plate formats and as such is compatible with screening applications. We agree with the reviewer that it is desirable to further showcase the operational advantages of the Gel-3D culture, either for mechanistic studies or translational applications. To this end, in the first submission we had sought to develop Gel-3D

cultures in multiwell plates to screen hPGCLC differentiation potentials of hiPSCs derived from eight non-obstructive azoospermia (NOA) patients. Our data show that hiPSCs from NOA participants retain the ability to give rise to hPGCLCs. Thus, we believe that our study has contained sufficient data to demonstrate the Gel-3D culture as a useful tool for hPGCLC induction and related mechanistic and translational studies. The new discussions have been added into the manuscript (Lines 245-256 on pages 12 & 13).

Specific comment (2): *The authors developed a co-culture system of hPSCs and AMLCs to investigate the requirement of paracrine signaling from AMLCs to PSCs. While this method is novel, the AMLC induction protocol employed was different from that used in the Gel-3D culture system. Could the authors clarify how the AMLCs induced using this protocol are similar to in vivo amnion or AMLCs induced using other protocols, such as in terms of gene expression profiles.?*

Response: We appreciate the reviewer for raising this point. Please see our response to Specific comment (4) from Reviewer #1.

Specific comment (3): *In addition, I think it is already known that BMP4, NODAL, and WNT signaling play important roles in PGC/PGCLC induction in humans and/or mice. Moreover, previous studies (e.g., Sasaki et al., 2016 Dev Cell) have demonstrated that the nascent amnion is the source of BMP4. Therefore, could the authors consider clarification on how the new method improves our understanding of the signaling mechanisms involved in hPGCLC induction?*

Response: We thank the reviewer for this comment. We agree with the reviewer that the involvements of BMP, WNT, and NODAL signaling in PGC/PGCLC specification have been established in the field. In this work, the investigations of the roles of these pathways are to characterize the Gel-3D culture and to understand the functional connections of these pathways with *ISL1*. As such, related data have only been included in Extended Data Figures. We have revised the main text to add these discussions (Lines 197-198 on page 10).

Specific comment (4): *The authors reported successful induction of hPGCLCs from NOS iPSCs using their new method. However, it would be valuable to understand how this new method has improved the induction of hPGCLCs from NOS iPSCs, specifically in terms of induction efficiency and quality. In this regard, could the authors clarify how the induction efficiency and/or quality of hPGCLCs was improved by their new method compared to a conventional method?*

Response: We thank the reviewer for this comment. Based on our results obtained from normal hPSC lines, the hPGCLC induction efficiency of the Gel-3D culture is comparable with that of existing approaches. It is challenging to evaluate the quality of hPGCLCs. Nonetheless, our comparative transcriptome analysis supports that hPGCLCs in the Gel-3D culture show transcriptome similarities with those in conventional hPGCLC induction protocols. We agree with the reviewer that it is desirable to further showcase the operational advantages of the Gel-3D culture, either for mechanistic studies or translational applications. To this end, in the first submission we had sought to develop Gel-3D cultures in multiwell plates to screen hPGCLC

differentiation potentials of hiPSCs derived from eight non-obstructive azoospermia (NOA) patients. Our data show that hiPSCs from NOA participants retain the ability to give rise to hPGCLCs. Thus, we believe that our study has contained sufficient data to demonstrate the Gel-3D culture as a useful tool for hPGCLC induction and related mechanistic and translational studies. The new discussions have been added into the manuscript (Lines 228-229 on page 11).

References:

- 1 Zheng, Y. *et al.* Controlled modelling of human epiblast and amnion development using stem cells. *Nature* **573**, 421-425 (2019).
- 2 Shao, Y. *et al.* Self-organized amniogenesis by human pluripotent stem cells in a biomimetic implantation-like niche. *Nature materials* **16**, 419 (2017).
- 3 Shao, Y. *et al.* A pluripotent stem cell-based model for post-implantation human amniotic sac development. *Nature communications* **8**, 1-15 (2017).
- 4 Resto Irizarry, A. M. *et al.* Machine learning-assisted imaging analysis of a human epiblast model. *Integrative Biology* **13**, 221-229 (2021).
- 5 Piccolo, F. M. *et al.* Role of YAP in early ectodermal specification and a Huntington's Disease model of human neurulation. *elife* **11**, e73075 (2022).
- 6 Terry, B. K. & Kim, S. The role of Hippo-YAP/TAZ signaling in brain development. *Developmental Dynamics* **251**, 1644-1665 (2022).
- 7 Davis, J. R. & Tapon, N. Hippo signalling during development. *Development* **146**, dev167106 (2019).
- 8 Neto, F. *et al.* YAP and TAZ regulate adherens junction dynamics and endothelial cell distribution during vascular development. *elife* **7**, e31037 (2018).

REVIEWER COMMENTS

Reviewer #1 (Remarks to the Author):

In this revised version of the manuscript, the authors have substantially improved the manuscript and addressed some of the comments of the reviewers. Having said this, some of the major criticism remains:

1. Novelty:

- The authors claim that their model is superior to the conventional EB-based protocol for hPGCLC specification for several reasons, including the fact that EBs are “not very convenient to implement and are incompatible with live imaging and thus imaging-based screening pipelines”. EBs have been used for more than 40 years, they are extremely easy to implement in any stem cell lab and are compatible with live imaging as shown for example in PMID 27530599. EBs are also compatible with screening applications as shown in PMID 34425190.

- The authors claim that Zheng et al, Nature, 2019 “did not draw conclusions or provide concrete evidence on the inductive role of AMLCs in hPGCLC specification”. This is not true. Zheng et al, Nature, 2019 used co-cultures of AMLCs and hPSCs to show that in the presence of AMLCs hPGCLCs become specified. This is exactly what they are showing again in this new manuscript.

As I mentioned in my previous comments the novelty of the paper relates to the identification of the amnion factor ISL1 as important for hPGCLC specification. The title does not reflect what is the novelty of the paper, as hPGCLCs have already been reported in embryonic-like cultures.

2. Co-culture experiments and inhibitor treatments: my initial concern was based on the fact that inhibitor treatments are added to a dish that contains two cell types, amniotic epithelial cells, and pluripotent cells. Therefore, it is not possible to determine whether the observed effects are cell-autonomous or non-cell-autonomous. In response to my comment, the authors claim that “the introduction of inhibitors should only interfere with paracrine interactions”. I think this is not correct. The inhibitor could for example affect autocrine WNT signalling in human ESCs, thereby compromising hPGCLC specification. The limitations of these experiments need to be acknowledged.

3. Rescue experiments: the authors now provide a rescue experiment, whereby they add exogenous BMP4 and/or WNT3A to the ISL1 KO 3D Gels or co-cultures. They conclude that there is an “effective rescue”. However, the co-culture experiments have not been quantified at all, and the quantification of the 3D Gel experiments only shows a potential partial rescue, which remains speculative as the data has not been statistically analyzed.

4. Defective WNT signalling downstream of ISL1: the authors now show decreased levels of WNT5A/5B/6 in ISL1 KO cells. WNT5A/5B are non-canonical WNT ligands. Given that hPGCLC specification is triggered by canonical WNT signaling, this experiment is not relevant to the paper and doesn't add additional support to the potential role of WNT signaling downstream of ISL1.

5. Quantifications for multiple experiments are still missing, despite my initial comment.

Reviewer #2 (Remarks to the Author):

I am happy with the response of the authors to my comments as well as to those of the other reviewers in which they address, in particular, issues of novelty. I also feel that the additions make the manuscript and its conclusions stronger.

Reviewer #3 (Remarks to the Author):

Although the authors added a few experiments to supplement their previous experimental results, the findings and novelties of the results are still fairly limited with ambiguous datasets.

1. Judging from the efficiency of obtaining early hPGCLCs, the methodology of so-called 3D culture has similar differentiation efficiency of hPGCLCs compared to others (including the one in Ref 6.) The developmental stage of hPGCLCs in this study is also similar to the early PGCs, but not further differentiated to more mature germ cells. If the authors could show that their 3D protocols have higher developmental potential to become functional or more mature germ cells (towards sperm or oocyte) than other protocols, perhaps this might help to claim that their protocol is more advanced or better mimicking the in vivo developmental pathway.

2. The majority of mechanistic studies of hPGCLC inductions by BMP, WNT, and NODAL are more like validations of previous publications cited in the paper and mentioned by reviewer 1. The only moderately novel result of this study is the potential induction of hPGCLCs by ISL1+ amniotic ectodermal cells in the neighboring sites of PGCLCs; however, the evidences of ISL1+ cell secreting BMP4 to induce PGC formation are ambiguous, indirect, and not solid. For example, only Fig. 3A show a few ISL1+ cells surrounding one or two faint SOX17+ stained cells (speculative because the DAPI did not colocalized with SOX17+). SOX17+ alone could be endodermal lineage at the early embryonic development and may

not be hPGCLCs. Moreover, the authors used TFAP2A instead of ISL1 to follow AMLCs in most of the staining experiments. Why did the authors change use the ectodermal marker to follow the ISL1+ AMLCs population? How would the authors know the TFAP2A cells developed from ISL1+ cells? AMCLs is a mixed population of ISL1+ and ISL1- cells, if ISL1+ really induced PGC lineage through BMP and WNT induction, one would expect a cluster of ISL1+ cells colocalized to a neighboring SOX17+TFRAP2C+NANOG+ hPGCLCs clusters. Could the authors show such evidence or similar clustering experiments?

3. The last set of experiments also seemed irrelevant or not necessary to support the claim of an efficient screening method for examining PGCLC differentiation potential of NOA samples. In Fig.4C, it seems that the percentage of hPGCLCs range from ~5% to 20%. The authors claimed that their wild-type hPGCLCs efficiency is around 20%, so is the difference really significant? What is the point of showing ethnicity distribution in Fig. 4d if there is only 8 samples and no difference of hPGCLCs among the patients?

Hence, I agree with Reviewer 1 that this manuscript would be more suitable for a specialized journal.

Reviewer #4 (Remarks to the Author):

I have thoroughly reviewed the authors' responses to my comments and the revised version of the manuscript. I found that all of my concerns have been appropriately addressed by the authors, and now I can recommend this work for publication in Nature Communications.

Response to Reviewer 1

In this revised version of the manuscript, the authors have substantially improved the manuscript and addressed some of the comments of the reviewers. Having said this, some of the major criticism remains:

Response: We thank the reviewer for his/her/their constructive comments, which have helped us improve the quality and impact of this work significantly.

Specific comment (1): Novelty:

- The authors claim that their model is superior to the conventional EB-based protocol for hPGCLC specification for several reasons, including the fact that EBs are “not very convenient to implement and are incompatible with live imaging and thus imaging-based screening pipelines”. EBs have been used for more than 40 years, they are extremely easy to implement in any stem cell lab and are compatible with live imaging as shown for example in PMID 27530599. EBs are also compatible with screening applications as shown in PMID 34425190.

Response: We thank the reviewer for this comment. Our statement in the last rebuttal letter could have been more accurate and clearer. We were not trying to criticize the general utility of EB for different biological and biomedical applications. In fact, we agree with the reviewer that EB provides a convenient experimental tool to study mammalian embryonic lineage development using stem cells, even in certain screening applications as suggested by the reviewer. Nonetheless, we hope the reviewer would agree with us that the conventional methods for inducing hPGCLCs using EB-based approaches leave a lot to be desired. In fact, the two publications mentioned by the reviewer showcase such limitations. Both publications use advanced two-photon microscopies for imaging and cell lineage reporter lines. Still the limited spatiotemporal resolutions and the fact that EB is free-floating present significant technical hurdles to obtain good-quality, quantitative data, particularly for rare cells, such as hPGCLCs. Such assays are also limited by the availability and quality of reporter cell lines. The above discussion has been added to the revised manuscript (Lines 233-240 on page 12).

- The authors claim that Zheng et al, Nature, 2019 “did not draw conclusions or provide concrete evidence on the inductive role of AMLCs in hPGCLC specification”. This is not true. Zheng et al, Nature, 2019 used co-cultures of AMLCs and hPSCs to show that in the presence of AMLCs hPGCLCs become specified. This is exactly what they are showing again in this new manuscript. As I mentioned in my previous comments the novelty of the paper relates to the identification of the amnion factor *ISL1* as important for hPGCLC specification. The title does not reflect what is the novelty of the paper, as hPGCLCs have already been reported in embryonic-like cultures.

Response: We agree with the reviewer and apologize for the confusion raised from our last rebuttal letter. We also thank the reviewer for recognizing the novelty of this work in identifying the role of amnion factor *ISL1* in hPGCLC specification. Since the most important contribution of this manuscript lies in the development of a new method for hPGCLC derivation from hPSCs and this new method is simple, effective and *in vivo*-relevant, we believe the title of the manuscript is a suitable choice.

Specific comment (2): Co-culture experiments and inhibitor treatments: my initial concern was based on the fact that inhibitor treatments are added to a dish that contains two cell types, amniotic epithelial cells, and pluripotent cells. Therefore, it is not possible to determine whether the observed effects are cell-autonomous or non-cell-autonomous. In response to my comment, the authors claim that “the introduction of inhibitors should only interfere with paracrine interactions”. I think this is not correct. The inhibitor could for example affect autocrine WNT signalling in human ESCs, thereby compromising hPGCLC specification. The limitations of these experiments need to be acknowledged.

Response: We thank the reviewer for the comment. Accordingly, we have revised the main text to acknowledge the limitation of our drug inhibition assays in the co-culture system (Lines 198-200 on page 10).

Specific comment (3): Rescue experiments: the authors now provide a rescue experiment, whereby they add exogenous BMP4 and/or WNT3A to the ISL1 KO 3D Gels or co-cultures. They conclude that there is an “effective rescue”. However, the co-culture experiments have not been quantified at all, and the quantification of the 3D Gel experiments only shows a potential partial rescue, which remains speculative as the data has not been statistically analyzed.

Response: We thank the reviewer for the comment. Accordingly, in this revised manuscript we conducted quantifications and statistical analyses for the rescue experiments (see **Extended Data Fig. 7f**). Main text of the manuscript has been updated accordingly (Line 183 on page 9).

Specific comment (4): Defective WNT signalling downstream of ISL1: the authors now show decreased levels of WNT5A/5B/6 in ISL1 KO cells. WNT5A/5B are non-canonical WNT ligands. Given that hPGLC specification is triggered by canonical WNT signaling, this experiment is not relevant to the paper and doesn’t add additional support to the potential role of WNT signaling downstream of ISL1.

Response: We thank the reviewer for this comment, and we agree! Accordingly, we have removed data related to WNT5A/5B/6 expression from **Fig. 3** and **Extended Data Fig. 7**. We have also modified main text accordingly.

Specific comment (5): Quantifications for multiple experiments are still missing, despite my initial comment.

Response: We thank the reviewer for this important comment. We have now added quantification data for co-culture experiments (see **Fig. 3d**, **Extended Data Fig. 6c&e** and **Extended Data Fig. 7f**). We have also revised main text accordingly.

Response to Reviewer 2

I am happy with the response of the authors to my comments as well as to those of the other reviewers in which they address, in particular, issues of novelty. I also feel that the additions make the manuscript and its conclusions stronger.

Response: We thank the reviewer for his/her/their insightful comments throughout the peer-review process, which have helped us improve the quality and impact of this work significantly.

Response to Reviewer 3

Although the authors added a few experiments to supplement their previous experimental results, the findings and novelties of the results are still fairly limited with ambiguous datasets.

1. Judging from the efficiency of obtaining early hPGCLCs, the methodology of so-called 3D culture has similar differentiation efficiency of hPGCLCs compared to others (including the one in Ref 6.) The developmental stage of hPGCLCs in this study is also similar to the early PGCs, but not further differentiated to more mature germ cells. If the authors could show that their 3D protocols have higher developmental potential to become functional or more mature germ cells (towards sperm or oocyte) than other protocols, perhaps this might help to claim that their protocol is more advanced or better mimicking the in vivo developmental pathway.

Response: We appreciate the reviewer's insightful suggestion. Nonetheless, developing a protocol to produce more mature or even functional germ cells remains a holy grail of the field and is clearly out of the scope of this current work.

2. The majority of mechanistic studies of hPGCLC inductions by BMP, WNT, and NODAL are more like validations of previous publications cited in the paper and mentioned by reviewer 1. The only moderately novel result of this study is the potential induction of hPGCLCs by ISL1+ amniotic ectodermal cells in the neighboring sites of PGCLCs; however, the evidences of ISL1+ cell secreting BMP4 to induce PGC formation are ambiguous, indirect, and not solid. For example, only Fig. 3A show a few ISL1+ cells surrounding one or two faint SOX17+ stained cells (speculative because the DAPI did not colocalized with SOX17+). SOX17+ alone could be endodermal lineage at the early embryonic development and may not be hPGCLCs. Moreover, the authors used TFAP2A instead of ISL1 to follow AMLCs in most of the staining experiments. Why did the authors change use the ectodermal marker to follow the ISL1+ AMLCs population? How would the authors know the TFAP2A cells developed from ISL1+ cells? AMCLs is a mixed population of ISL1+ and ISL1- cells, if ISL1+ really induced PGC lineage through BMP and WNT induction, one would expect a cluster of ISL1+ cells colocalized to a neighboring SOX17+TFRAP2C+NANOG+ hPGCLCs clusters. Could the authors show such evidence or similar clustering experiments?

Response: We appreciate the reviewer's comment. Yes, it is true that SOX17 is also an endodermal lineage marker. However, based on our scRNA-seq data, there is no endodermal lineage in our system, and hPGCLCs are the only possible cells that can express SOX17 in our system. To clarify this point, we have revised main text accordingly (see Lines 145-148 on page 8). About using TFAP2A to track amniotic cell development, as far as we know, TFAP2A shows upregulated expression in early amniotic cells and remain expressed in amniotic cells. Thus, TFAP2A is a good marker to use for tracking amniotic cell differentiation. As a matter of fact, ISL1+ amnion cells are developed from TFAP2A+ cells. To address the reviewer's concern about image quality, we have added a set of new images to **Fig. 3a**. As can be seen clearly from these images, clusters of SOX17+NANOG+ hPGCLCs are in close proximity to ISL1+ amniotic cells. We have also revised main text accordingly (Lines 148-149 on page 8).

3. The last set of experiments also seemed irrelevant or not necessary to support the claim of an efficient screening method for examining PGCLC differentiation potential of NOA samples.

Response: We thank the reviewer for this comment. The purpose of conducting experiments with patient cell lines is to showcase the great potential of the Gel-3D culture system to be widely adopted by the research community for hPGCLC induction and translational studies and demonstrate that our protocol can be applied to both established hPSC lines and patient-derived hiPSC lines in a high-throughput fashion, with all different cell lines assayed in parallel. To address this comment, we have added these new discussions in the revised manuscript (Lines 244-250 on page 12).

In Fig. 4C, it seems that the percentage of hPGCLCs range from ~5% to 20%. The authors claimed that their wild-type hPGCLCs efficiency is around 20%, so is the difference really significant?

Response: We apologize for the confusion here. To clarify, the efficiency of hPGCLC induction from wild-type hPSC lines is about 10% on day 3 and 20% on day 8. Data presented for NOA cell lines in **Fig. 4C** pertain to hPGCLC induction efficiency on day 3. To avoid this confusion, we have added some clarifications in the caption of **Fig. 4**.

What is the point of showing ethnicity distribution in Fig. 4d if there is only 8 samples and no difference of hPGCLCs among the patients?

Response: We thank the reviewer for this comment. The inclusion of ethnicity distribution in **Fig. 4d** serves to highlight our commitment to equal experiment with cell lines from diverse ethnic backgrounds (see recent discussions about genetic diversity in hPSC research, *Nature Communications* **13**, 7301, 2022). By showing this diversity, we aim to emphasize the broad applicability and inclusivity of our research. It's essential to note that the objective of this figure is not to compare hPGCLC specification across different ethnicities, but rather to demonstrate our efforts in ensuring that our research encompasses a wide range of genetic backgrounds. The above new discussion has been incorporated into the revised manuscript (Lines 254-255 on page 13).

Hence, I agree with Reviewer 1 that this manuscript would be more suitable for a specialized journal.

Response to Reviewer 4

I have thoroughly reviewed the authors' responses to my comments and the revised version of the manuscript. I found that all of my concerns have been appropriately addressed by the authors, and now I can recommend this work for publication in Nature Communications.

Response: We thank the reviewer for his/her/their insightful comments throughout the peer-review process, which have helped us improve the quality and impact of this work significantly.

REVIEWERS' COMMENTS

Reviewer #1 (Remarks to the Author):

In this revised version of the manuscript, the authors have addressed the technical issues that I raised. Having said this, I still have the same reservations as reviewer 3: (1) the described method is not superior to already published methods and (2) the molecular insight that we gain compared to previously published studies is limited.